# BAYESIAN LOW-RANK ADAPTATION FOR LARGE LANGUAGE MODELS

**Adam X. Yang**[1*]    **Maxime Robeyns**[1*]    **Xi Wang**[2]    **Laurence Aitchison**[1]
[1]University of Bristol    [2]University of Massachusetts, Amherst
{adam.yang,maxime.robeyns.2018,laurence.aitchison}@bristol.ac.uk
xwang3@cs.umass.edu

## ABSTRACT

Low-rank adaptation (LoRA) has emerged as a new paradigm for cost-efficient fine-tuning of large language models (LLMs). However, fine-tuned LLMs often become overconfident especially when fine-tuned on small datasets. Bayesian methods, with their inherent ability to estimate uncertainty, serve as potent tools to mitigate overconfidence and enhance calibration. In this work, we introduce Laplace-LoRA, which applies a Bayesian approach to the LoRA parameters. Specifically, Laplace-LoRA applies a Laplace approximation to the posterior over the LoRA parameters, considerably improving the calibration of fine-tuned LLMs.

## 1 INTRODUCTION

In recent years, fine-tuning large language models (LLMs) have become increasingly important (Houlsby et al., 2019; Hu et al., 2021; Liu et al., 2022; Ding et al., 2022; 2023). Fine-tuning is used both to adapt LLMs for specific tasks and to create general instruction-following models (e.g. using Reinforcement Learning from Human Feedback; RLHF Wei et al., 2021; Ouyang et al., 2022; Chung et al., 2022; Wang et al., 2022).

However, fine-tuned LLMs have a notable limitation: they often exhibit overconfidence (Jiang et al., 2021; Xiao et al., 2022; He et al., 2023; Tian et al., 2023; OpenAI, 2023). This is particularly problematic in safety-critical applications or when making decisions in areas where limited data is available, such as medical diagnosis, finance and experimental design (Singhal et al., 2022; Wu et al., 2023; Lampinen et al., 2023; Li et al., 2022). Consequently, there is an urgent need for strategies that enhance the calibration of fine-tuned LLMs, ensuring that their predictions are as trustworthy as they are powerful.

Bayesian deep learning is commonly proposed as a solution to overconfidence in deep networks (e.g. Blundell et al., 2015; Zhang et al., 2019; Kristiadi et al., 2020; Ober & Aitchison, 2021; Fortuin et al., 2021; Aitchison et al., 2021). Historically, the field of Bayesian deep learning has frequently considered ResNets for image classification (Shridhar et al., 2019; Dusenberry et al., 2020; Izmailov et al., 2021). While there are a few previous papers on Bayesian language models (e.g. Tran et al., 2019; Xue et al., 2021; Fan et al., 2020; Cinquin et al., 2021; Zhang et al., 2019; Chen & Li, 2023), most of them are focused on language model pre-training (though see Related Work for exceptions). These methods have not yet seen widespread adoption, potentially because they are costly to scale up to larger models, and large-scale pre-trained models are already reasonably well-calibrated (OpenAI, 2023).

Importantly, the advantages of Bayesian fine-tuning are much clearer than the advantages of Bayesian pretraining. First, fine-tuned models are typically poorly calibrated (Jiang et al., 2021; Xiao et al., 2022; He et al., 2023; Tian et al., 2023), even after very large-scale instruction fine-tuning (OpenAI, 2023), and Bayes might help us improve this calibration. Second, this poor calibration may arise in fine-tuning settings, as there is often far less data available in fine-tuning than pre-training. Bayes might help us to reason about uncertainty given these limited datasets. Finally, the computational burden of fine-tuning is typically much smaller than that of pre-training, implying more headroom for any additional computational costs that might arise as a result of Bayesian inference.

---

*Equal contribution.

However, LLMs can be very large. Indeed, fine-tuning all the weights of an LLM is often prohibitively costly, in which case Bayesian fine-tuning will of course also not be possible. Instead, a number of more efficient approaches have emerged under the banner of parameter-efficient fine-tuning (PEFT Liu et al., 2022; Ding et al., 2022; 2023; Shi & Lipani, 2023). PEFT methods tend to train a small number of additional parameters on top of the underlying fixed-pre-trained LLM. For instance, a common approach is to fine-tune only low-rank adapters for each weight matrix (LoRA; (Hu et al., 2021)). PEFT approaches such as LoRA have democratized LLM fine-tuning and are implemented in a number of widely adopted libraries including OpenDelta (Ding et al., 2023) and PEFT (Mangrulkar et al., 2022).

Importantly, PEFT approaches such as LoRA should also enable more efficient Bayesian finetuning. We develop the first Bayesian inference method designed specifically for the LoRA parameters in LLM fine-tuning. Specifically, we use a post-hoc Laplace approximation (MacKay, 1992; Ritter et al., 2018; Daxberger et al., 2021b;a; Antorán et al., 2022), which allows us to keep the standard, highly efficient pre-training and fine-tuning pipelines *exactly* the same. We then use Laplace to compute the uncertainties over just the small number of LoRA (Hu et al., 2021) parameters. We call the resulting method Laplace-LoRA, and we show dramatic improvements in calibration on fine-tuned LlaMA2-7B on six common sense reasoning tasks and in out-of-distribution settings.

## 2  RELATED WORK

Past work on integrating Bayesian inference with language models has usually operated in the large-scale pre-training setting (Tran et al., 2019; Xue et al., 2021; Cinquin et al., 2021; Zhang et al., 2019; Chen & Li, 2023), where the advantages of Bayes are unclear, because pre-training datasets are very large and large-scale pretrained models seem to be reasonably well-calibrated even without Bayes (Kadavath et al., 2022; OpenAI, 2023). In contrast, our work operates in the fine-tuning setting, where the advantages of Bayes are far more evident, e.g. because even large-scale instruction fine-tuning gives poor calibration (OpenAI, 2023).

While Fan et al. (2020) and Zhang et al. (2021) do consider fine-tuning, they make the unusual choice to define a prior and approximate posterior over the attention weights, rather than over parameters (i.e. weight matrices). Our method — Laplace LoRA — has two key benefits over their approach. First, Laplace LoRA is a post-hoc method that keeps the pretraining and fine-tuning process *exactly* the same. Therefore Laplace LoRA is able to exploit very efficient pre-existing implementations of these methods (Ding et al., 2022; Mangrulkar et al., 2022; Dettmers et al., 2023), which is not possible when defining priors and approximate posteriors over attention weights. Second, Laplace LoRA massively reduces the dimensionality of Bayesian inference problem, either as against full finetuning where we would need to reason about the posterior over all weights, or as against Fan et al. (2020) and Zhang et al. (2021), which must reason about the posterior over attention weights at every layer for every fine-tuning datapoint. There are 32 attention heads per layer and 32 layers in LLaMA 7B, if we assume 1000 sequences of length 30 in the finetuning set, that gives 32 layers $\times 32$ heads per layer $\times 30^2$ attention weights per head $\times$ 30 sequences $\approx$ one billion attention weights. That one billion attention weights contrasts to only around 6 million LoRA parameters in our finetuning.

Laplace inference for Bayesian neural networks is well-studied (Ritter et al., 2018; Kristiadi et al., 2020; Immer et al., 2021; Antorán et al., 2022; Daxberger et al., 2021a; Deng et al., 2022). However, the only application of Laplace approximations to language models that we know of comes from Daxberger et al. (2021a) which used DistillBERT (a 66 million parameter model) as a feature-extractor, and applied Laplace inference only at the final linear last-layer. In contrast, in our work we consider Laplace inference at all layers in models 100 times larger, by using LoRA adapters.

A separate line of research has been dedicated to regularizing language model fine-tuning to improve calibration. For instance, Mixout (Lee et al., 2019) stochastically substitutes model weights with their pre-trained counterparts. Park & Caragea (2022) adapted Mixup (Zhang et al., 2017) to augment the dataset during language model fine-tuning. Meanwhile, He et al. (2023) introduced a KL regularization between the output distributions of both fine-tuned and pre-trained language models, and added an $L_2$ regularization to the final embedding vector. However, their regularization relies on the masked language model objective which only applies to BERT-like models. Critically, this body of work is orthogonal to our work in that they give better settings for the fine-tuned weights. Laplace-LoRA does not change the fine-tuned weights at all, so it could use weights from standard

fine-tuning, or it could use the improved weights given by these methods. Instead, Laplace-LoRA uses a Laplace approximation to estimate uncertainty around any given mean value of the weights. Laplace-LoRA thus has the potential to improve calibration for *any* good method for generating fine-tuned weights.

# 3 BACKGROUND

## 3.1 LOW-RANK ADAPTATION (LORA)

LLMs have a large number of large weight matrices, denoted $\mathbf{W}_0 \in \mathbb{R}^{n_{\text{out}} \times n_{\text{in}}}$, with inputs $\mathbf{a}$ and outputs $\mathbf{h}$. In LoRA (Hu et al., 2021), we keep $\mathbf{W}_0$ fixed, and introduce a perturbation to the weight matrix, $\Delta \mathbf{W}$,

$$\mathbf{h} = \mathbf{W}_0 \mathbf{a} + \Delta \mathbf{W} \mathbf{a} = \mathbf{W}_0 \mathbf{a} + \mathbf{B} \mathbf{A} \mathbf{a}. \tag{1}$$

Critically, $\Delta \mathbf{W}$ is low-rank as it is written as the product of two matrices, $\mathbf{B} \in \mathbb{R}^{n_{\text{out}} \times n_{\text{lr}}}$ and $\mathbf{A} \in \mathbb{R}^{n_{\text{lr}} \times n_{\text{in}}}$ where $n_{\text{lr}}$ is significantly smaller than $n_{\text{in}}$ or $n_{\text{out}}$ (e.g. 4096), for instance, we use $n_{\text{lr}} = 8$. Therefore, the total number of LoRA parameters for this weight matrix is $n_{\text{lr}}(n_{\text{in}} + n_{\text{out}})$, which is typically far smaller than the underlying number of parameters in the full matrix, $n_{\text{in}} n_{\text{out}}$. LoRA has shown great success in fine-tuning large scale models efficiently, and can be adapted to language or vision architectures (Mangrulkar et al., 2022). Note that one of the key motivations for introducing LoRA for LLM finetuning was the huge memory cost of maintaining the average gradient and average squared gradients of the e.g. 7 billion parameters in the optimizer. This multiplies the memory required by a factor of 3, relative to the memory required to just load the weights. LoRA massively reduces this memory cost to only 3 times the number of parameters in the LoRA adapters, as it only optimizes the LoRA adapters.

## 3.2 LAPLACE APPROXIMATIONS

In Bayesian inference for classification or next token prediction, the goal is to find the full posterior,

$$\mathrm{P}\left(\boldsymbol{\theta}|\mathbf{X}, \mathbf{y}\right) \propto \mathrm{P}\left(\mathbf{y}|\mathbf{X}, \boldsymbol{\theta}\right) \mathrm{P}\left(\boldsymbol{\theta}\right). \tag{2}$$

Here, $\mathbf{X} \in \mathcal{T}^{N \times S}$ is the input, where $\mathcal{T}$ is the set of possible tokens, $N$ is the number of sequences, $S$ is the (max) sequence length. The targets are denoted $\mathbf{y} \in \mathcal{Y}^N$, where $\mathcal{Y}$ could be different from $\mathcal{T}$ (e.g. in sentiment classification) or could be the same as $\mathcal{T}$ (e.g. for next token prediction). Further, $\mathrm{P}\left(\boldsymbol{\theta}|\mathbf{X}, \mathbf{y}\right)$ is the posterior, $\mathrm{P}\left(\mathbf{y}|\boldsymbol{\theta}, \mathbf{X}\right)$ is the likelihood (e.g. softmax Categorical distribution for classification tasks). We use an isotropic Gaussian prior, with precision $\lambda$,

$$\mathrm{P}\left(\boldsymbol{\theta}\right) = \mathcal{N}(\mathbf{0}, \lambda^{-1} \mathbf{I}). \tag{3}$$

Calculating this posterior is usually intractable. The Laplace approximation begins by finding the maximum a-posteriori (MAP) solution (MacKay, 1992) (i.e. the maximum of the log-joint, $\mathcal{L}(\mathbf{y}, \mathbf{X}; \boldsymbol{\theta})$),

$$\mathcal{L}(\mathbf{y}, \mathbf{X}; \boldsymbol{\theta}) = \log \mathrm{P}\left(\mathbf{y}|\mathbf{X}, \boldsymbol{\theta}\right) + \log \mathrm{P}\left(\boldsymbol{\theta}\right) = \log \mathrm{P}\left(\boldsymbol{\theta}|\mathbf{X}, \mathbf{y}\right) + \text{const} \tag{4}$$

$$\boldsymbol{\theta}_{\text{MAP}} = \underset{\boldsymbol{\theta}}{\operatorname{argmax}} \, \mathcal{L}(\mathbf{y}, \mathbf{X}; \boldsymbol{\theta}). \tag{5}$$

Then the Laplace approximation consists of a second-order Taylor expansion of the log-joint around $\boldsymbol{\theta}_{\text{MAP}}$,

$$\mathcal{L}(\mathbf{y}, \mathbf{X}; \boldsymbol{\theta}) \approx \mathcal{L}(\mathbf{y}, \mathbf{X}; \boldsymbol{\theta}_{\text{MAP}}) - \frac{1}{2}(\boldsymbol{\theta} - \boldsymbol{\theta}_{\text{MAP}})^T (\nabla_{\boldsymbol{\theta}}^2 \mathcal{L}(\mathbf{y}, \mathbf{X}; \boldsymbol{\theta})|_{\boldsymbol{\theta}_{\text{MAP}}})(\boldsymbol{\theta} - \boldsymbol{\theta}_{\text{MAP}}). \tag{6}$$

Since the log-joint is now a quadratic function of $\boldsymbol{\theta}$, the approximate posterior becomes a Gaussian centered at $\boldsymbol{\theta}_{\text{MAP}}$ with covariance given by the inverse of the Hessian,

$$\mathrm{P}\left(\boldsymbol{\theta}|\mathcal{D}\right) \approx \mathcal{N}\left(\boldsymbol{\theta}; \boldsymbol{\theta}_{\text{MAP}}, \boldsymbol{\Sigma}\right), \tag{7}$$

$$\boldsymbol{\Sigma} = -(\nabla_{\boldsymbol{\theta}}^2 \mathcal{L}(\mathbf{y}, \mathbf{X}; \boldsymbol{\theta})|_{\boldsymbol{\theta}_{\text{MAP}}})^{-1} = -(\nabla_{\boldsymbol{\theta}}^2 \log \mathrm{P}\left(\mathbf{y}|\mathbf{X}, \boldsymbol{\theta}\right)|_{\boldsymbol{\theta}_{\text{MAP}}} + \lambda \mathbf{I})^{-1}. \tag{8}$$

To ensure positive definiteness of the covariance, the general approach is to transform gradients into estimates of the Hessian using either the Fisher information or the Generalized Gauss Newton (GGN) matrix (see Kunstner et al., 2019, for further details). We use the Fisher information,

$$\mathbf{F}(\boldsymbol{\theta}) = \sum_{n=1}^{N} \mathbb{E}_{\mathrm{P}(y|f_{\boldsymbol{\theta}}(\mathbf{x}_n))} \left[ \nabla_{\boldsymbol{\theta}} \, \mathrm{P} \left( y|f_{\boldsymbol{\theta}}(\mathbf{x}_n) \right) \left( \nabla_{\boldsymbol{\theta}} \, \mathrm{P} \left( y|f_{\boldsymbol{\theta}}(\mathbf{x}_n) \right) \right)^T \right], \tag{9}$$

where the expectation is taken over the model's output distribution.

Importantly, the full Hessian or Fisher is a $P \times P$ matrix, where $P$ is the number of parameters. For LLMs, this might translate to a matrix with dimensions of 7 billion by 7 billion, which is clearly intractable. Even if we consider the Hessian only over the low-rank adapters this matrix is still too large; for example, applying rank 8 LoRA on Llama2-7B still yields roughly 6 million trainable parameters. Consequently, we follow the usual Laplace approximation approach by imposing further structure on the Hessian. In particular, we consider either just the Hessian for the last-layer, or using Kronecker-factored (KFAC) structure for individual weight matrices (Ritter et al., 2018; Daxberger et al., 2021a). In KFAC, we approximate the Fisher using blocks for each linear layer. For the $\ell$th linear layer, we compute the block by denoting the input as $\mathbf{a}_\ell$ and the output as $\mathbf{b}_\ell$. Then, the Fisher is,

$$\mathbf{F}_\ell = \sum_{n=1}^{N} \mathbb{E}_{\mathrm{P}(y|f_{\boldsymbol{\theta}}(\mathbf{x}_n))} \left[ (\mathbf{a}_{\ell-1}\mathbf{a}_{\ell-1}^T) \otimes (\mathbf{g}_\ell \mathbf{g}_\ell^T) \right]. \tag{10}$$

where $\mathbf{g}_\ell = \nabla_{\mathbf{b}_\ell} \log \mathrm{P}\left(\mathbf{y}|\mathbf{X}, \boldsymbol{\theta}\right)$ is the gradient of the the log-likelihood gradient with respect to the outputs.

Using Laplace approximations has strong connections to linearizing the neural network (Kunstner et al., 2019; Immer et al., 2021). As such, it is commonly found that predicting under the linearized model is more effective than e.g. sampling the approximate posterior over weights (Foong et al., 2019; Daxberger et al., 2021a; Deng et al., 2022; Antorán et al., 2022), (see Appendix A for further details). In particular,

$$f_{\boldsymbol{\theta}}(\mathbf{x}_*) \approx f_{\boldsymbol{\theta}_{\mathrm{MAP}}}(\mathbf{x}_*) + \nabla_{\boldsymbol{\theta}} f_{\boldsymbol{\theta}}(\mathbf{x}_*)|_{\boldsymbol{\theta}_{\mathrm{MAP}}}^T (\boldsymbol{\theta} - \boldsymbol{\theta}_{\mathrm{MAP}}). \tag{11}$$

where $\mathbf{x}_*$ is a test-input. This approach is also known as the linearized Laplace approximation.

Since we have the approximated posterior in Eq. (7) and the linearized model in Eq. (11), we can integrate out the posterior on weights and get a Gaussian posterior on output logits,

$$f_{\boldsymbol{\theta}}(\mathbf{x}_*) \sim \mathcal{N}\left(f_{\boldsymbol{\theta}_{\mathrm{MAP}}}(\mathbf{x}_*), \boldsymbol{\Lambda}\right), \tag{12}$$

where

$$\boldsymbol{\Lambda} = (\nabla_{\boldsymbol{\theta}} f_{\boldsymbol{\theta}}(\mathbf{x}_*)|_{\boldsymbol{\theta}_{\mathrm{MAP}}}^T) \boldsymbol{\Sigma} (\nabla_{\boldsymbol{\theta}} f_{\boldsymbol{\theta}}(\mathbf{x}_*)|_{\boldsymbol{\theta}_{\mathrm{MAP}}}). \tag{13}$$

Subsequently, we can optimize the prior precision $\lambda$ using the closed form Laplace marginal likelihood (model evidence) (Immer et al., 2021; Daxberger et al., 2021a) on the training dataset,

$$\mathrm{P}\left(\mathbf{y}|\mathbf{X}\right) = \int \mathrm{P}\left(\mathbf{y}|\mathbf{X}, \boldsymbol{\theta}\right) \mathrm{P}\left(\boldsymbol{\theta}\right) d\boldsymbol{\theta} \approx \exp(\mathcal{L}(\mathbf{y}, \mathbf{X}; \boldsymbol{\theta}_{\mathrm{MAP}}))(2\pi)^{D/2}|\boldsymbol{\Sigma}|^{1/2}, \tag{14}$$

where we have applied the Taylor approximation in Eq. (6). Crucially, unlike other post-hoc calibration methods, post-hoc Laplace does not require a separate validation set. This feature is particularly beneficial for small-scale datasets where training data is scarce. To obtain samples of $f_{\boldsymbol{\theta}}(\mathbf{x}_*)$, we can decompose the covariance using the Cholesky factorization, $\boldsymbol{\Lambda} = \mathbf{L}\mathbf{L}^T$,

$$\tilde{f}_{\boldsymbol{\theta}}(\mathbf{x}_*) = f_{\boldsymbol{\theta}_{\mathrm{MAP}}}(\mathbf{x}_*) + \mathbf{L}\boldsymbol{\xi}, \tag{15}$$

where $\boldsymbol{\xi}$ is a vector of IID standard normal random variables. We compute the Bayesian model average by computing the average probabilities (passing the sampled logits through softmax function) under the Gaussian random noise from $\boldsymbol{\xi}$. There are common approaches to approximating the Bayesian model average arising from Eq. (15), including the generalized probit approximation and the Laplace bridge (Daxberger et al., 2021a; Kristiadi et al., 2020; Lu et al., 2020; MacKay, 1998), but we find that they perform considerably worse than naive Monte-Carlo sampling, potentially due to their ignorance of covariance terms (Appendix A).

## 4 METHODS

Laplace-LoRA uses post-hoc Laplace Kronecker-factored approximations for the LoRA adapters. Specifically, we treat the adapter $\mathbf{B}\mathbf{A}\mathbf{x}$ in Eq. (1) as two separate linear layers with weights $\mathbf{A} \in \mathbb{R}^{n_{\text{lr}} \times n_{\text{in}}}$ and $\mathbf{B} \in \mathbb{R}^{n_{\text{out}} \times n_{\text{lr}}}$ respectively rather than as a single linear layer with a low rank weight matrix. One issue with this approach is that the LoRA adapters are $n_{\text{lora}} \times d$ or $d \times n_{\text{lora}}$ dimensional, so while one Kronecker-factor is small ($n_{\text{lora}} \times n_{\text{lora}}$), the other is large ($d \times d$, where $d = 4096$ in LlaMA2-7B attention layers). These $d \times d$ Kronecker factors are roughly the same size as the underlying weight matrices, so representing them explicitly would eliminate any memory benefits of working with LoRA adapters. As such, we are forced to use a low-rank representation of this Kronecker factor. Specifically, we use a rank-$n_{\text{kfac}}$ representation (the rank of this low-rank approximation to the Kronecker factor will in general be different from the rank of the LoRA adapters).

To ensure that we retain memory efficiency at all steps, we need to be able to:

1. Compute the low-rank approximation to the Kronecker factor "incrementally" (i.e. all the computations are low-rank and we do not e.g. compute the full-rank factor first, and then find a low-rank form; Appendix E.1).

2. Optimize the marginal likelihood using the low-rank approximation (Appendix E.2).

3. Low-rank linearized prediction (Appendix E.3).

## 5 RESULTS

We consider post-hoc Laplace approximations applied to LoRA parameters (Laplace-LoRA) at model checkpoints $\boldsymbol{\theta}_{\text{MAP}}$ obtained from standard fine-tuning. In particular, we used the PEFT library (Mangrulkar et al., 2022) for fine-tuning LlaMA2-7B (Touvron et al., 2023b) on common-sense reasoning tasks (multiple choice or True/False classification). We applied LoRA to queries, values, and the output layer, using default hyperparameters from HuggingFace (Wolf et al., 2020; Mangrulkar et al., 2022) (see Appendix C). The fine-tuning was carried out with a batch size of 4 for 10,000 iterations. For True/False or multiple choice questions we selected the next token logits corresponding to True/False or A/B/C/D depending on each dataset (refer to Appendix B for our prompt templates), and fine-tuned the LLM to maximize the likelihood of the correct token.

Model checkpoints were saved every 1000 gradient steps, and at each checkpoint, we applied post-hoc Laplace-LoRA with KFAC; we predicted using the linearized model as described in the Methods section. For clarity, we refer to the full Laplace-LoRA approximation (which is applied to all LoRA fine-tuned weights) as LA, and the last-layer Laplace-LoRA approximation (targeting only the LoRA weights at the output layer to emulate standard last-layer Laplace approximation (Kristiadi et al., 2020; Daxberger et al., 2021a) ) as LLLA.

We assessed the efficacy of Laplace-LoRA by evaluating the negative log-likelihood and expected calibration error of LlaMA2-7B during fine-tuning on common-sense reasoning tasks. These metrics are further discussed in Appendix D. We also drew comparisons with established baselines including Monte-Carlo dropout (Gal & Ghahramani, 2016) with an ensemble size of 10 (with a dropout rate of 0.1 during fine-tuning), checkpoint ensemble (Chen et al., 2017) with an ensemble size of 3 using the most recent 3 checkpoints when available, deep ensemble (Lakshminarayanan et al., 2017; Wang et al., 2023; Zhai et al., 2023) with three LoRA fine-tuned LLMs, and temperature scaling (Guo et al., 2017). Since achieving good calibration and accurate uncertainty estimation is significantly more challenging on small-scale datasets (Zhao et al., 2020), our focus lies primarily on datasets comprising fewer than 10,000 training examples. We use the publicly available validation split across all benchmarks as the test set to evaluate performances at checkpoints.

### 5.1 IN-DISTRIBUTION FINE-TUNING AND EVALUATION

We began our evaluation with in-distribution fine-tuning on the following common sense reasoning tasks: Winogrande-small (WG-S), Winogrande-medium (WG-M) (Sakaguchi et al., 2021), ARC-Challenge (ARC-C), ARC-Easy (ARC-E) (Clark et al., 2018), openbook QA (OBQA) (Mihaylov

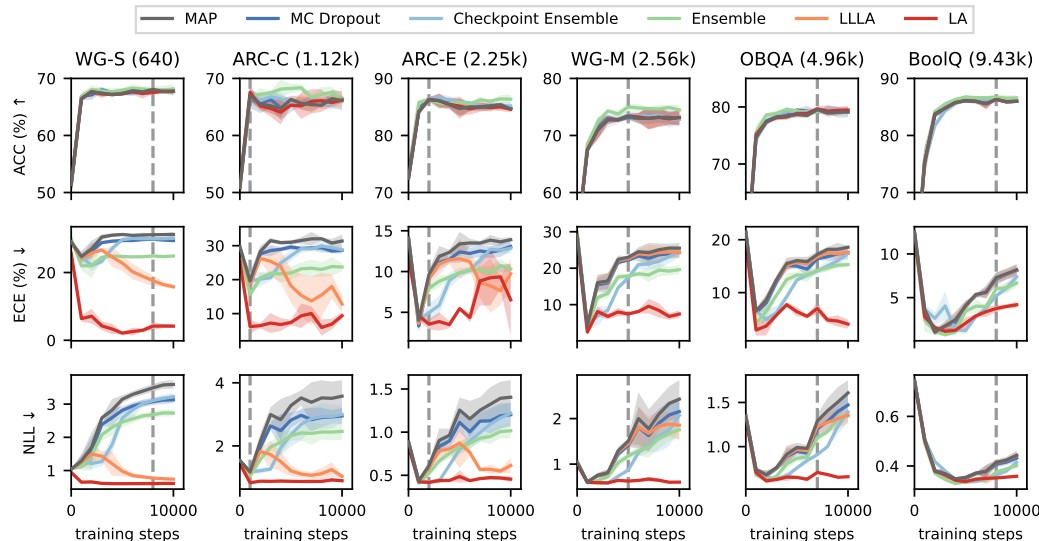

Figure 1: Fine-tuning of LlaMA2-7B across six common sense reasoning tasks (presented column-wise, with number of training examples in brackets), evaluated on the test set every 1000 gradient steps. The vertical dashed line gives the number of training steps with optimal MAP performance, and indicates that Laplace is likely to offer benefits even when combined with early stopping. MAP: standard fine-tuning; MC Dropout: Monte-Carlo dropout; Checkpoint Ensemble: ensembling three most recent checkpoints; Ensemble: ensembling three LoRA fine-tuned models; LLLA: last-layer Laplace approximation on LoRA weights in the output layer; LA: full Laplace approximation on all LoRA weights.

et al., 2018), and BoolQ (Clark et al., 2019) benchmarks[1]. One of the key advantages of Laplace approximations is that they give an approximation of the model evidence which can be used to tune hyperparameters on the training set alone, without needing a separate validation set. Thus, we first considered a setting with only a training set and a test set, with no distinct validation set for hyperparameter tuning. For our Laplace-LoRA methods (LA and LLLA), we optimized the prior variances using the Laplace model evidence (detailed in Algorithm 2 in Appendix F).

Figure 1 presents the accuracy (ACC), expected calibration error (ECE) and negative log-likelihood (NLL) on the evaluation set for each fine-tuned checkpoint. Since all other methods maintained similar ACC compared to MAP, our primary concern revolved around ECE and NLL. Notably, the very large NLLs for standard (MAP) fine-tuning as training proceeds indicates that overconfidence remains an important issue when when fine-tuning Llama-7B. Monte-Carlo dropout offers marginal improvements over MAP, while an checkpoint ensemble offers slightly more gains at the beginning. LoRA ensemble sometimes offer an improvement in ACC, but its improvements on ECE and NLL are always limited. Conversely, LLLA seems to give substantial improvements on some datasets (WG-S, ARC-C, ARC-E, but little if any improvement on others WG-M, OBQA, BoolQ). In contrast, LA consistently offers dramatic benefits in terms of ECE and NLL, indicating overconfidence on the evaluation set is dramatically reduced.

Additionally, we present these results in tabular form in Table 1. We stopped at 5000 steps to somewhat mirror the effect of early stopping (note however that choosing the correct point to stop is difficult if not impossible to determine in this setting which lacks a validation set.) LA consistently offered considerable gains over MAP in terms of ECE and NLL, achieving best NLL among all methods across all datasets, while maintaining similar ACC. Note that while this table gives only the performance at one time point, Figure 1 shows that LA usually offers considerable benefits in calibration across all points in training. Additionally, in Fig. 1, we highlight the point with highest *test* accuracy with the dashed line (as this is the highest test accuracy, it cannot practically be used to early-stop, but it does indicate that were we to optimally early-stop, we would still get dramatic benefits in calibration from LA.)

---

[1]We truncate the context in BoolQ for computational considerations (see Appendix C).

Table 1: Comparison of different post-hoc methods applied to the fine-tuned LlaMA2-7B across six common sense reasoning tasks. Results are evaluated at the early stopping point of 5000 gradient steps. We report standard deviations in subscripts, and bold numbers that are statistical significant.

| Metrics | Methods | WG-S | ARC-C | ARC-E | WG-M | OBQA | BoolQ |
|---------|---------|------|-------|-------|------|------|-------|
| ACC ↑ | MAP | $67.4_{0.3}$ | $66.3_{0.6}$ | $84.7_{1.5}$ | $73.4_{0.4}$ | $78.7_{0.4}$ | $86.1_{0.2}$ |
| | MC Drop | $67.8_{0.1}$ | $65.3_{1.0}$ | $85.0_{1.3}$ | $73.2_{0.5}$ | $79.5_{0.2}$ | $86.0_{0.3}$ |
| | Ckpt Ens | $67.4_{0.2}$ | $65.5_{0.4}$ | $85.8_{0.2}$ | $73.6_{0.7}$ | $79.1_{0.1}$ | $86.3_{0.2}$ |
| | Ensemble | $68.0_{0.3}$ | $68.2_{0.7}$ | $85.8_{0.5}$ | $75.0_{0.5}$ | $79.3_{0.4}$ | $86.8_{0.1}$ |
| | LLLA | $67.4_{0.3}$ | $66.2_{0.4}$ | $84.7_{1.5}$ | $73.4_{0.4}$ | $78.7_{0.4}$ | $86.1_{0.2}$ |
| | LA | $67.3_{0.2}$ | $65.3_{0.2}$ | $85.1_{1.5}$ | $73.4_{0.3}$ | $78.9_{0.2}$ | $86.1_{0.2}$ |
| ECE ↓ | MAP | $31.2_{0.3}$ | $31.0_{0.5}$ | $13.4_{1.3}$ | $23.0_{0.1}$ | $16.1_{0.6}$ | $4.0_{0.5}$ |
| | MC Drop | $29.4_{0.3}$ | $29.6_{0.8}$ | $12.4_{1.2}$ | $22.2_{0.2}$ | $15.0_{0.4}$ | $4.1_{0.4}$ |
| | Ckpt Ens | $29.7_{0.6}$ | $27.0_{1.5}$ | $9.8_{0.6}$ | $17.4_{0.9}$ | $12.4_{0.3}$ | $\mathbf{1.2_{0.4}}$ |
| | Ensemble | $24.7_{0.3}$ | $21.9_{1.7}$ | $9.9_{0.2}$ | $17.9_{0.6}$ | $13.3_{0.6}$ | $3.5_{0.2}$ |
| | LLLA | $22.8_{2.0}$ | $18.2_{4.4}$ | $11.6_{2.2}$ | $22.6_{0.2}$ | $15.8_{0.6}$ | $4.0_{0.5}$ |
| | LA | $\mathbf{2.1_{0.3}}$ | $\mathbf{7.4_{0.7}}$ | $\mathbf{5.4_{0.2}}$ | $\mathbf{7.4_{0.4}}$ | $\mathbf{6.4_{0.8}}$ | $2.1_{0.6}$ |
| NLL ↓ | MAP | $3.15_{0.10}$ | $3.28_{0.29}$ | $1.26_{0.13}$ | $1.51_{0.05}$ | $0.99_{0.05}$ | $0.35_{0.01}$ |
| | MC Drop | $2.81_{0.11}$ | $2.82_{0.21}$ | $1.11_{0.10}$ | $1.41_{0.03}$ | $0.95_{0.04}$ | $0.35_{0.01}$ |
| | Ckpt Ens | $2.58_{0.15}$ | $2.36_{0.34}$ | $0.80_{0.06}$ | $0.87_{0.06}$ | $0.76_{0.01}$ | $\mathbf{0.33_{0.00}}$ |
| | Ensemble | $2.46_{0.14}$ | $2.32_{0.14}$ | $0.83_{0.06}$ | $1.18_{0.05}$ | $0.87_{0.03}$ | $0.34_{0.00}$ |
| | LLLA | $0.98_{0.13}$ | $1.21_{0.16}$ | $0.87_{0.26}$ | $1.45_{0.06}$ | $0.97_{0.04}$ | $0.35_{0.01}$ |
| | LA | $\mathbf{0.60_{0.01}}$ | $\mathbf{0.88_{0.03}}$ | $\mathbf{0.49_{0.06}}$ | $\mathbf{0.63_{0.02}}$ | $\mathbf{0.65_{0.01}}$ | $0.34_{0.01}$ |

Next, we considered a more standard setting, in which the training set was partitioned into an 80% subset for training and a 20% validation subset. Notably, this usually led to a slight decrease in performance, as we were no longer training on all the data (Table 1 against Table 2). This allowed us to benchmark against a widely-adopted post-hoc calibration technique, temperature scaling (Guo et al., 2017), which requires a separate validation set to tune the temperature hyperparameter. Additionally, it is possible to tune the prior precision in Laplace using the NLL on the validation set, rather than the "more Bayesian" approach of tuning prior precision using the Laplace estimate of the model evidence. The standard approach is to perform grid search to maximize validation log-likelihood (Daxberger et al., 2021a). However, fine-grained grid search is costly. Instead, utilizing our reparametrized model average (Eq. 15), weperformed stochastic gradient descent for the precision hyperparameters, using the validation NLL as the objective (see Algorithm 3 in Appendix F). Table 2 shows the result of tuning temperature and Laplace prior on a validation set at the checkpoint with the best MAP validation accuracy. Although temperature scaling is very competitive, LA still gives slightly improved NLL and usually improves ECE.

## 5.2 EVALUATIONS UNDER DISTRIBUTION SHIFT

Finally, since real world deployment of LLMs requires them to generalize to data coming from a different domain compared to the fine-tuning dataset (Ouyang et al., 2022; Touvron et al., 2023a;b), we conduct further experiments to evaluate fine-tuned models under distribution shift using the model checkpoint fine-tuned on OBQA in Table 2. Our evaluations under distribution shift are divided into two groups, smaller distribution shift and larger distribution shift. For smaller distribution shift, we use ARC datasets ARC-C and ARC-E, as they also contain common sense reasoning multiple choice questions similar to those in OBQA. For larger distribution shift, we pick four subjects from the MMLU task (Hendrycks et al., 2020): computer science (CS), engineering (Eng), law, and health, where each subject contains one or more tasks. A detailed breakdown is available in Table 19 in Appendix I.

Table 3 shows the OOD evaluation results using the early-stopped checkpoint on OBQA from Table 1. LA clearly provides substantial improvements over all other methods in terms of Expected Calibration Error (ECE) and Negative Log-Likelihood (NLL) while maintaining similar accuracy (ACC), with a slight improvement in the Computer Science (CS) subject. Table 8 shows the evaluation results under distribution shift using the best validation accuracy checkpoint on OBQA from Table 2. Once again,

Table 2: Comparison of different post-hoc methods applied to the fine-tuned LlaMA2-7B across six common sense reasoning tasks, with a validation set split from the training set used for tuning temperature and Laplace prior precision. Results are evaluated at the best MAP performance checkpoint observed on the validation set.

| Metrics | Methods | WG-S | ARC-C | ARC-E | WG-M | OBQA | BoolQ |
|---|---|---|---|---|---|---|---|
| ACC ↑ | MAP | $67.0_{0.6}$ | $64.9_{1.1}$ | $85.2_{0.6}$ | $73.7_{0.9}$ | $77.7_{0.8}$ | $85.8_{0.4}$ |
| | MC Drop | $66.7_{0.3}$ | $64.9_{1.9}$ | $85.1_{0.5}$ | $73.5_{0.9}$ | $77.7_{0.2}$ | $85.9_{0.4}$ |
| | Ckpt Ens | $66.7_{0.3}$ | $64.9_{1.1}$ | $85.2_{0.6}$ | $73.8_{1.0}$ | $78.2_{0.2}$ | $85.4_{0.3}$ |
| | Temp | $67.0_{0.6}$ | $64.9_{1.1}$ | $85.2_{0.6}$ | $73.7_{0.9}$ | $77.7_{0.8}$ | $85.8_{0.4}$ |
| | LLLA | $66.9_{0.5}$ | $66.1_{0.6}$ | $84.8_{0.5}$ | $73.7_{0.9}$ | $77.6_{0.7}$ | $85.8_{0.4}$ |
| | LA | $66.9_{0.6}$ | $66.9_{1.1}$ | $85.4_{0.4}$ | $73.7_{1.0}$ | $78.1_{0.7}$ | $85.8_{0.4}$ |
| ECE ↓ | MAP | $30.8_{1.8}$ | $26.1_{1.4}$ | $8.9_{0.3}$ | $24.9_{1.3}$ | $9.8_{1.0}$ | $7.4_{0.1}$ |
| | MC Drop | $29.5_{1.6}$ | $25.6_{0.7}$ | $8.8_{0.6}$ | $23.5_{1.2}$ | $8.8_{0.8}$ | $7.5_{0.1}$ |
| | Ckpt Ens | $25.2_{1.6}$ | $26.1_{1.4}$ | $8.9_{0.3}$ | $22.8_{1.4}$ | $4.7_{0.5}$ | $3.2_{0.5}$ |
| | Temp | $12.8_{0.9}$ | $\mathbf{4.6_{1.0}}$ | $4.7_{0.8}$ | $6.3_{1.6}$ | $7.2_{2.6}$ | $2.5_{0.3}$ |
| | LLLA | $11.6_{1.3}$ | $5.6_{2.1}$ | $4.2_{0.3}$ | $\mathbf{3.8_{1.4}}$ | $5.4_{0.4}$ | $\mathbf{1.7_{0.5}}$ |
| | LA | $\mathbf{7.8_{1.9}}$ | $7.5_{1.2}$ | $\mathbf{3.4_{0.8}}$ | $4.8_{1.6}$ | $\mathbf{3.5_{0.4}}$ | $1.9_{0.3}$ |
| NLL ↓ | MAP | $2.75_{0.57}$ | $1.64_{0.19}$ | $0.54_{0.03}$ | $2.43_{0.50}$ | $0.71_{0.03}$ | $0.43_{0.01}$ |
| | MC Drop | $2.54_{0.49}$ | $1.55_{0.16}$ | $0.52_{0.04}$ | $2.12_{0.35}$ | $0.71_{0.04}$ | $0.43_{0.01}$ |
| | Ckpt Ens | $1.31_{0.04}$ | $1.64_{0.18}$ | $0.54_{0.03}$ | $1.89_{0.24}$ | $0.65_{0.02}$ | $0.35_{0.01}$ |
| | Temp | $0.68_{0.01}$ | $0.90_{0.01}$ | $0.43_{0.02}$ | $0.58_{0.01}$ | $0.67_{0.02}$ | $0.35_{0.00}$ |
| | LLLA | $0.68_{0.01}$ | $0.94_{0.02}$ | $0.44_{0.01}$ | $0.56_{0.01}$ | $0.66_{0.02}$ | $0.35_{0.00}$ |
| | LA | $\mathbf{0.66_{0.02}}$ | $\mathbf{0.86_{0.02}}$ | $\mathbf{0.41_{0.02}}$ | $\mathbf{0.55_{0.01}}$ | $\mathbf{0.62_{0.01}}$ | $\mathbf{0.34_{0.00}}$ |

Table 3: Comparison of different post-hoc methods across six out-of-distribution datasets. Results are evaluated using the fine-tuned LlaMA2-7B on OBQA as in Table 1.

| Metrics | Methods | ID | Smaller Distribution Shift | | Larger Distribution Shift | | | |
|---|---|---|---|---|---|---|---|---|
| | | OBQA | ARC-C | ARC-E | CS | Eng | Law | Health |
| ACC ↑ | MAP | $78.7_{0.4}$ | $67.9_{1.4}$ | $77.7_{0.3}$ | $42.0_{3.2}$ | $41.2_{2.0}$ | $37.4_{0.4}$ | $48.3_{0.3}$ |
| | MC Drop | $79.5_{0.2}$ | $67.7_{0.6}$ | $77.2_{0.6}$ | $41.9_{2.2}$ | $39.6_{1.7}$ | $37.9_{0.4}$ | $48.2_{0.9}$ |
| | Ckpt Ens | $79.1_{0.2}$ | $67.9_{0.8}$ | $77.4_{0.8}$ | $41.1_{2.0}$ | $38.7_{1.2}$ | $37.7_{0.2}$ | $48.2_{0.6}$ |
| | Ensemble | $68.0_{0.3}$ | $69.0_{0.7}$ | $77.0_{1.1}$ | $41.9_{1.7}$ | $41.9_{2.8}$ | $38.0_{0.3}$ | $48.4_{0.5}$ |
| | LLLA | $78.7_{0.4}$ | $68.1_{0.0}$ | $78.1_{0.0}$ | $45.6_{0.0}$ | $38.9_{0.0}$ | $37.1_{0.0}$ | $48.5_{0.0}$ |
| | LA | $78.9_{0.2}$ | $69.2_{0.0}$ | $78.5_{0.0}$ | $45.1_{0.0}$ | $39.1_{0.0}$ | $37.3_{0.0}$ | $49.1_{0.0}$ |
| ECE ↓ | MAP | $16.1_{0.6}$ | $22.2_{1.2}$ | $15.8_{1.0}$ | $34.2_{3.1}$ | $38.4_{1.7}$ | $35.2_{0.7}$ | $34.2_{0.8}$ |
| | MC Drop | $15.0_{0.4}$ | $21.4_{0.4}$ | $15.5_{1.0}$ | $33.7_{2.0}$ | $38.6_{2.9}$ | $34.2_{0.6}$ | $33.5_{0.2}$ |
| | Ckpt Ens | $10.1_{0.3}$ | $17.7_{0.7}$ | $12.1_{0.6}$ | $29.1_{2.3}$ | $32.5_{1.8}$ | $32.1_{0.1}$ | $29.0_{0.3}$ |
| | Ensemble | $24.7_{0.3}$ | $18.5_{1.0}$ | $13.4_{0.7}$ | $30.4_{2.6}$ | $31.7_{1.3}$ | $32.7_{0.4}$ | $31.0_{0.9}$ |
| | LLLA | $15.8_{0.6}$ | $21.3_{0.0}$ | $14.8_{0.0}$ | $30.3_{0.0}$ | $39.7_{0.0}$ | $33.6_{0.0}$ | $33.5_{0.0}$ |
| | LA | $\mathbf{6.4_{0.8}}$ | $\mathbf{8.8_{0.0}}$ | $\mathbf{6.2_{0.0}}$ | $\mathbf{15.8_{0.0}}$ | $\mathbf{25.5_{0.0}}$ | $\mathbf{24.7_{0.0}}$ | $\mathbf{17.9_{0.0}}$ |
| NLL ↓ | MAP | $0.99_{0.05}$ | $1.30_{0.07}$ | $1.04_{0.10}$ | $1.90_{0.12}$ | $2.19_{0.15}$ | $2.12_{0.03}$ | $2.09_{0.08}$ |
| | MC Drop | $0.95_{0.04}$ | $1.24_{0.06}$ | $1.01_{0.09}$ | $1.86_{0.10}$ | $2.14_{0.13}$ | $2.09_{0.02}$ | $2.05_{0.07}$ |
| | Ckpt Ens | $0.68_{0.03}$ | $1.03_{0.03}$ | $0.80_{0.03}$ | $1.55_{0.04}$ | $1.72_{0.01}$ | $1.94_{0.01}$ | $1.74_{0.02}$ |
| | Ensemble | $2.46_{0.14}$ | $1.15_{0.07}$ | $0.91_{0.04}$ | $1.74_{0.10}$ | $1.95_{0.10}$ | $2.04_{0.03}$ | $1.95_{0.11}$ |
| | LLLA | $0.97_{0.04}$ | $1.30_{0.00}$ | $0.99_{0.00}$ | $1.80_{0.00}$ | $2.18_{0.00}$ | $2.06_{0.00}$ | $2.05_{0.00}$ |
| | LA | $\mathbf{0.65_{0.01}}$ | $\mathbf{0.90_{0.00}}$ | $\mathbf{0.70_{0.00}}$ | $\mathbf{1.35_{0.00}}$ | $\mathbf{1.58_{0.00}}$ | $\mathbf{1.74_{0.00}}$ | $\mathbf{1.50_{0.00}}$ |

both LLLA and LA typically yield improvements in ECE and NLL over other methods, indicating their effectiveness in out-of-distribution scenarios.

In addition to experiments on LlaMA in the main text, we also present results for fine-tuning encoder-based language models RoBERTa-base and RoBERTa-large (Liu et al., 2019) on text classification tasks in Appendix G.2 Notably, Monte-Carlo dropout performs much better on RoBERTa-base but is less effective on RoBERTa-large; checkpoint ensemble often performs much worse than MAP in contrast to its performance on LlaMA; LLLA still offers tiny improvements over MAP. Besides, we have also added additional experiments on fine-tuning Mistral-7B (Jiang et al., 2023) in Appendix G.4. Overall, full Laplace-LoRA (LA) provides consistent and significant improvements in uncertainty estimation across models (RoBERTa, LlaMA, Mistral) and tasks (text classification, common sense

Table 4: Comparison of running time and memory cost of standard LoRA finetuning for 10,000 iterations vs standard LoRA finetuning for 10,000 along with accumulating low-rank Kronecker factors for Laplace every 1000 epochs. Experiements carried out using Llama2 7B, LoRA denotes standard LoRA fine-tuning for 10,000 steps, Laplace-LoRA denotes LoRA fine-tuning for 10,000 steps along with post-hoc Laplace approximation on all LoRA weights.

| Metric | Method | WG-S | WG-M | ARC-C | ARC-E | OBQA | BoolQ |
|---|---|---|---|---|---|---|---|
| Time (Seconds) ↓ | LoRA | 8287 | 8684 | 10500 | 10301 | 9268 | 16964 |
| | LoRA + Laplace | 8442 | 8839 | 11141 | 10942 | 10526 | 17636 |
| Memory (GB) ↓ | LoRA | 8.34 | 8.33 | 8.37 | 8.43 | 8.59 | 10.48 |
| | LoRA + Laplace | 8.43 | 8.43 | 8.73 | 8.75 | 8.84 | 10.93 |

reasoning). Supplementary experiments using diagonal Fisher as a Hessian approximation instead of KFAC are documented in Appendix G.6. However, we found that diagonal Laplace approximations do not offer consistent improvements in ECE and NLL, emphasizing the importance of using KFAC to model correlations between weights in Laplace-LoRA. We have open sourced our code implementations (Appendix J).

## 5.3 MEMORY AND RUNTIME COST

The primary advantages of LoRA lie in its memory and computational efficiency. In this section, we present a comparative analysis of the memory and time costs associated with LoRA fine-tuning alone, and LoRA fine-tuning followed by post-hoc application of Laplace-LoRA. As illustrated in Table 4, implementing Laplace-LoRA incurs only an additional around 1–5% memory overhead. Note that in the experiments in Table 4, there is very little overhead in terms of time (about 10%), despite the fact that we actually accumulate the low-rank Kronecker factors 10 times (once at the end of every 1000 iterations, and we ran these experiments for 10000 iterations total). In practice, we would only need to accumulate low-rank Kronecker factors once, so the overhead should be around 1%. This empirical analysis demonstrates that our low-rank KFAC approximation of Laplace-LoRA effectively maintains LoRA's efficiency.

## 6 CONCLUSION

In this work, we proposed Laplace-LoRA, for Bayesian parameter-efficient fine-tuning of LLMs. Our method is scalable to larger networks due to its focus on the LoRA weights. Additionally, by applying Laplace approximations post-hoc, after fine-tuning, we require no changes to efficient implementations of the standard fine-tuning process. In our experiments, we observed significant gains in expected calibration error and negative log-likelihood, indicating an improved estimation of uncertainty. Our approach underpins the potential of Laplace-LoRA as a step towards the development of more reliable and trustworthy LLMs.

## 7 ACKNOWLEDGEMENTS

This work was carried out using the computational facilities of the Advanced Computing Research Centre, University of Bristol - http://www.bris.ac.uk/acrc/. We would like to thank Dr Stewart for funding compute resources used in this project, and we would like to thank Yixuan Su for helpful discussions.

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

# A    LAPLACE PREDICTIVE POSTERIOR APPROXIMATIONS

In this section, we review several methods for obtaining the predictive posterior of Laplace approximation.

## A.1    MONTE CARLO SAMPLING

As descibed in main text, we can obtain a closed-form Gaussian posterior on output logits,

$$f_{\boldsymbol{\theta}}(\mathbf{x}_*) \sim \mathcal{N}\left(f_{\boldsymbol{\theta}_{\mathrm{MAP}}}(\mathbf{x}_*), \boldsymbol{\Lambda}\right), \tag{16}$$

where

$$\boldsymbol{\Lambda} = (\nabla_{\boldsymbol{\theta}} f_{\boldsymbol{\theta}}(\mathbf{x}_*)|_{\boldsymbol{\theta}_{\mathrm{MAP}}}^T)\boldsymbol{\Sigma}(\nabla_{\boldsymbol{\theta}} f_{\boldsymbol{\theta}}(\mathbf{x}_*)|_{\boldsymbol{\theta}_{\mathrm{MAP}}}). \tag{17}$$

To obtain samples of $f_{\boldsymbol{\theta}}(\mathbf{x}_*)$, we can decompose the covariance using the Cholesky factorization $\boldsymbol{\Lambda} = \mathbf{L}\mathbf{L}^T$ with

$$\tilde{f}_{\boldsymbol{\theta}}(\mathbf{x}_*) = f_{\boldsymbol{\theta}_{\mathrm{MAP}}}(\mathbf{x}_*) + \mathbf{L}\boldsymbol{\xi}, \tag{18}$$

where $\boldsymbol{\xi}$ is a vector of IID standard normal random variables. We can compute the Bayesian model average by computing the average probabilities (passing the sampled logits through softmax function) under the Gaussian random noise from $\boldsymbol{\xi}$.

## A.2    PROBIT APPROXIMATION

A closed-form approximation of the predictive posterior for classification can be obtained by integrating out the posterior over weights with a generalized probit approximation (Lu et al., 2020; Daxberger et al., 2021a) of the likelihood,

$$p(y_*|\mathbf{x}_*, \mathcal{D}) \approx \mathrm{Categorical}\left(y_*, \mathrm{softmax}\left(\frac{f_{\boldsymbol{\theta}_{\mathrm{MAP}}}(\mathbf{x}_*)}{\sqrt{1 + \frac{\pi}{8} \operatorname{diag}(\boldsymbol{\Lambda})}}\right)\right), \tag{19}$$

where

$$\boldsymbol{\Lambda} = (\nabla_{\boldsymbol{\theta}} f_{\boldsymbol{\theta}}(\mathbf{x}_*)|_{\boldsymbol{\theta}_{\mathrm{MAP}}}^T)\boldsymbol{\Sigma}(\nabla_{\boldsymbol{\theta}} f_{\boldsymbol{\theta}}(\mathbf{x}_*)|_{\boldsymbol{\theta}_{\mathrm{MAP}}}). \tag{20}$$

Although probit approximation provably preserves decision boundary in binary sigmoid classification (Kristiadi et al., 2020), it does hold for softmax multiclass classification.

## A.3    LAPLACE BRIDGE

The Laplace bridge (MacKay, 1998; Daxberger et al., 2021a) maps a Gaussian $\mathcal{N}(\boldsymbol{\mu}, \boldsymbol{\Sigma})$ to a Dirichlet distribution $\mathcal{D}(\boldsymbol{\alpha})$ over classes with

$$\boldsymbol{\alpha}_i = \frac{1}{\boldsymbol{\Sigma}_{ii}}\left(1 - \frac{2}{C} + \frac{\exp(\boldsymbol{\mu}_i)}{C^2}\sum_j \exp(-\boldsymbol{\mu}_j)\right), \tag{21}$$

where $C$ denotes the number of classes. Similar to the generalized probit approximation, it also ignores the covariance terms and only considers the diagonal of the covariance $\boldsymbol{\Sigma}_{ii}$.

## A.4    EMPIRICAL COMPARISON

Figure 2 shows a comparison of these approximations in fine-tuning Llama2-7B and applying full Laplace-LoRA post-hoc. Specifically, we considered Monte Carlo sampling using the full covariance (MC joint), Monte Carlo sampling only using the diagonal covariance (MC indep), generalized probit approximation (probit), and Laplace bridge (bridge). MC joint consistently achieves highest accuracy and among the best NLL, while bridge is often the worst, probit and MC indep can sometimes give suboptimal performance. This is likely due to bridge, probit and MC indep are all approximations that ignored the covariances between logits, whereas MC joint faithfully approximates the true predictive posterior.

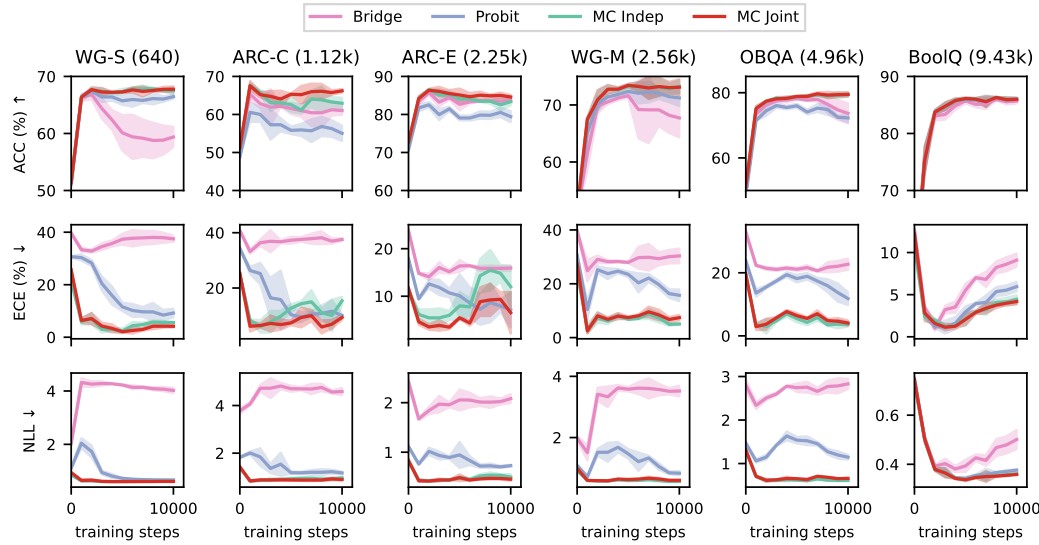

Figure 2: Fine-tuning of LlaMA2-7B across six common sense reasoning tasks, comparing different Laplace predictive posterior approximations: Laplace bridge approximation (bridge), generalized probit approximation (probit), Monte Carlo sampling using the diagonal covariance (MC indep), and Monte Carlo sampling using the full covariance (MC joint).

| Task | Prompt |
|------|--------|
| Winogrande (WG-S/WG-M) | Select one of the choices that answers the following question: {question} Choices: A. {option1}. B {option2}. Answer: |
| ARC (ARC-C/ARC-E) | Select one of the choices that answers the following question: {question} Choices: A. {choice1}. B. {choice2}. C. {choice2}. D. {choice2}. Answer: |
| Openbook QA (OBQA) | Select one of the choices that answers the following question: {question} Choices: A. {choice1}. B. {choice2}. C. {choice2}. D. {choice2}. Answer: |
| BoolQ | Answer the question with only True or False: {question} Context: {passage}. |

Table 5: Prompt templates for fine-tuning LlaMA-7B on common sense reasoning tasks.

## B  PROMPT TEMPLATES FOR LLAMA2 COMMON SENSE REASONING TASKS

We present our prompt templates used to fine-tune LlaMA2 on common sense reasoning tasks in Table 5. For ARC datasets, although the majority of quesstions have four choices, there are a tiny amount of questions with three or five choices which we remove for consistency. In ARC-C, there are 1/1119 with three choices and 1/1119 with five choices in the training set; 3/299 with three choices and 1/299 with five choices in the evaluation set. In ARC-E, there are 6/2251 with three choices and 4/2251 with five choices in the trainng set; 1/570 with three choices and 2/570 with five choices in the evaluation set.

## C  HYPERPARAMETERS

We follow the default hyperparameters from Huggingface's Transformer (Wolf et al., 2020) and PEFT (Mangrulkar et al., 2022) libraries. Hyperparameters used for fine-tuning RoBERTa-base and RoBERTa-large with LoRA are shown in Table 6, those for fine-tuning LlaMA-7B are shown in Table 7. Note the only differences between the two settings are a smaller batch size to fit into our GPU memory, and a longer max sequence length to account for prompt length.

| Hyperparameter | Value |
|---|---|
| LoRA $r$ | 8 |
| LoRA $\alpha$ | 16 |
| Dropout Probability | 0.1 |
| Weight Decay | 0 |
| Learning Rate | $5 \times 10^{-5}$ |
| Learning Rate Scheduler | Linear |
| Batch Size | 32 |
| Max Sequence Length | 256 |

Table 6: Hyperparameters used in fine-tuning RoBERTa-base and RoBERTa-large with LoRA.

| Hyperparameter | Value |
|---|---|
| LoRA $r$ | 8 |
| LoRA $\alpha$ | 16 |
| Dropout Probability | 0.1 |
| Weight Decay | 0 |
| Learning Rate | $5 \times 10^{-5}$ |
| Learning Rate Scheduler | Linear |
| Batch Size | 4 |
| Max Sequence Length | 300 |

Table 7: Hyperparameters used in fine-tuning LlaMA-7B with LoRA.

## D  METRICS FOR UNCERTAINTY QUANTIFICATION

There are two commonly used metrics for measuring uncertainty quantification in neural networks: negative log-likelihood (NLL) and expected calibration error (ECE). NLL computes the sum of negative expected log probability of predicting the true label,

$$\text{NLL} = \sum_{i=1}^{N} -\log P(\hat{y}_i = y_i), \tag{22}$$

where $P(\hat{y}_i)$ is the model's output distribution, $y_i$ is the true label. NLL is also equivalent to cross-entropy between the one-hot data distribution and the model's output distribution. NLL encourages the model to give high probability to correct answers. If the model is overconfident in a wrong answer, then the probability of giving the right answer would be low which raises NLL. On the other hand, ECE measures the alignment between model's confidence and accuracy, by binning the highest predicted probabilities and compute a weighted average between the difference of average accuracy and confidence in each bin,

$$\text{ECE} = \sum_{m=1}^{M} \frac{|B_m|}{N} |\text{acc}(B_m) - \text{conf}(B_m)|, \tag{23}$$

where $\text{acc}(B_m)$ and $\text{conf}(B_m)$ are the average accuracy and average confidence in each bin,

$$\text{acc}(B_m) = \frac{1}{|B_m|} \sum_{i \in B_m} \mathbf{1}(\hat{y}_i = y_i), \qquad \text{conf}(B_m) = \frac{1}{|B_m|} \sum_{i \in B_m} P(\hat{y}_i), \tag{24}$$

and $|B_m|$ is the number of examples in bin $m$. However, expected calibration error cannot be optimized directly like negative log-likelihood, as a completely random model will have the same accuracy and confidence for each datapoint, thus achieving zero ECE (Ashukha et al., 2020).

## E  LOW-RANK COMPUTATIONS

The usual K-FAC approach is to write,

$$\mathbf{\Sigma}_{\text{like}} = \mathbf{\Sigma}_{\text{like; a}} \otimes \mathbf{\Sigma}_{\text{like; b}} \tag{25}$$

As this is for the LoRA adapters, there's one small and one large covariance matrix. Without loss of generality, assume that $\mathbf{\Sigma}_{\text{like; a}} \in \mathbb{R}^{n_{\text{lora}} \times n_{\text{lora}}}$ is the small matrix, so we represent it as full-rank, and we assume that $\mathbf{\Sigma}_{\text{like; b}} \in \mathbb{R}^{d \times d}$ is the large matrix, so we represent it as a low-rank,

$$\mathbf{\Sigma}_{\text{like; b}}^{-1} \approx \mathbf{B}\mathbf{B}^T. \tag{26}$$

Thus, we write $\mathbf{\Sigma}_{\text{like; b}}$ in low-rank form, in terms of $\mathbf{B} \in \mathbb{R}^{d \times n_{\text{kfac}}}$.

The posterior precision is the sum of the likelihood and prior precisions,

$$\mathbf{\Sigma}_{\text{post}}^{-1} = \mathbf{\Sigma}_{\text{like; a}}^{-1} \otimes \mathbf{\Sigma}_{\text{like; b}}^{-1} + \tfrac{1}{\sigma^2}\mathbf{I} \otimes \mathbf{I}. \tag{27}$$

Substituting our low-rank approximation for the large Kronecker factor,

$$\mathbf{\Sigma}_{\text{post}}^{-1} = \mathbf{\Sigma}_{\text{like; a}}^{-1} \otimes \left(\mathbf{B}\mathbf{B}^T\right) + \tfrac{1}{\sigma^2}\mathbf{I} \otimes \mathbf{I}. \tag{28}$$

Using the mixed product property, and setting $\mathbf{\Sigma}_{\text{like; a}}^{-1} = \mathbf{L}\mathbf{L}^T$ (e.g. $\mathbf{L}$ is the Cholesky decomposition of $\mathbf{\Sigma}_{\text{like}}^{-1}$),

$$\mathbf{\Sigma}_{\text{post}}^{-1} = (\mathbf{L} \otimes \mathbf{B})(\mathbf{L} \otimes \mathbf{B})^T + \tfrac{1}{\sigma^2}\mathbf{I} \otimes \mathbf{I}. \tag{29}$$

This is in the form of a diagonal matrix, $\frac{1}{\sigma^2}\mathbf{I} \otimes \mathbf{I}$ plus a low-rank matrix, $(\mathbf{L} \otimes \mathbf{B})(\mathbf{L} \otimes \mathbf{B})^T$. Note that the rank of the low-rank matrix here is $n_{\text{lora}} n_{\text{kfac}}$. We're going to compute everything we need using this low-rank form.

### E.1 INCREMENTAL COMPUTATION OF LOW-RANK FACTOR

The first question is how to estimate the low-rank components, without ever computing a large $d \times d$ matrix. The answer — an incremental low-rank SVD — is given in Alg. 1.

---

**Algorithm 1** Memory efficient estimate of low-rank $\mathbf{B}$ such that $\mathbf{B}\mathbf{B}^T \approx \sum_{t=1}^{T} \mathbf{b}_t \mathbf{b}_t^T$.

---

Initialize to $0$
$\mathbf{B} = \mathbf{0}$
**for** $t = 1, ..., T$ **do**
    Combine current low-rank estimate, $\mathbf{B}$, with new activities, $\mathbf{b}_t = \mathbf{a}_t$, or new gradients, $\mathbf{b}_t = \mathbf{g}_t$
    Could combine with several vectors, if available
    $\mathbf{B}' = (\mathbf{B} \quad \mathbf{b}_t)$
    New low-rank estimate from top $n_{\text{kfac}}$ components of SVD
    $\mathbf{V}$ doesn't matter as its just a rotation
    $\mathbf{U}, \mathbf{S}, \mathbf{V}^T = \text{svd}(\mathbf{B}')$
    $\mathbf{B} \leftarrow \mathbf{U}[:, : n_{\text{kfac}}]\mathbf{S}[: n_{\text{kfac}}, : n_{\text{kfac}}]$
**end for**

---

### E.2 MARGINAL LIKELIHOOD OPTIMIZATION WITH THE MATRIX DETERMINANT LEMMA

To optimize the marginal likelihood (Eq.), we need the log-determinant of the posterior. To compute this log-determinant efficient in the low-rank setting, we use the matrix determinant lemma,

$$\log\det\left(\mathbf{\Sigma}_{\text{post}}^{-1}\right) = \log\det\left(\tfrac{1}{\sigma^2}\mathbf{I} \otimes \mathbf{I}\right) + \log\det\left(\mathbf{I} + \sigma^2 (\mathbf{L} \otimes \mathbf{B})^T (\mathbf{L} \otimes \mathbf{B})\right) \tag{30}$$

Applying the mixed product property,

$$\log\det\left(\mathbf{\Sigma}_{\text{post}}^{-1}\right) = -n_{\text{lora}} d \log\sigma^2 + \log\det\left(\mathbf{M}\right) \tag{31}$$

where,

$$\mathbf{M} = \Big(\mathbf{I}_{n_{\text{lora}} n_{\text{kfac}}} + \sigma^2 \underbrace{\mathbf{L}^T\mathbf{L}}_{n_{\text{lora}} \times n_{\text{lora}}} \otimes \underbrace{\mathbf{B}^T\mathbf{B}}_{n_{\text{kfac}} \times n_{\text{kfac}}}\Big) \tag{32}$$

$$\underbrace{\phantom{XXXXXXXXXXXXXXX}}_{n_{\text{lora}} n_{\text{kfac}} \times n_{\text{lora}} n_{\text{kfac}}}$$

So we can explicitly compute $\mathbf{L}^T\mathbf{L} \otimes \mathbf{B}^T\mathbf{B}$ and hence $\mathbf{M} \in \mathbb{R}^{n_{\text{lora}} n_{\text{kfac}} \times n_{\text{lora}} n_{\text{kfac}}}$ as long as $n_{\text{lora}} n_{\text{kfac}}$ is not too big. In our case, $n_{\text{lora}} = 4$ and $n_{\text{kfac}} = 10$, so $n_{\text{lora}} n_{\text{kfac}} = 40$.

---

**Algorithm 2** Optimize Laplace prior precision using the training set model evidence

---

Initialize prior precision $\lambda$, learning rate $\eta$, optimization steps $M$
Obtain $\theta_{\text{MAP}}$ from a fine-tuned checkpoint, pre-compute $\log P(\mathbf{y}|\mathbf{X}, \boldsymbol{\theta})$ and Fisher $\mathbf{F}$
**for** $step = 1, ..., M$ **do**
    Compute posterior covariance $\boldsymbol{\Sigma} = \mathbf{F} + \lambda \mathbf{I}$
    Calculate $\mathcal{L}(\mathbf{y}, \mathbf{X}; \boldsymbol{\theta}) = \log P(\mathbf{y}|\mathbf{X}, \boldsymbol{\theta}) + \log P(\boldsymbol{\theta}) = \log P(\mathbf{y}|\mathbf{X}, \boldsymbol{\theta}) + \lambda \|\boldsymbol{\theta}_{\text{MAP}}\|_2^2$
    Perform a gradient step with respect to $\lambda$ to maximize log model evidence (Eq.14):
        $\lambda \leftarrow \lambda + \eta \nabla_\lambda \left( \mathcal{L}(\mathbf{y}, \mathbf{X}; \boldsymbol{\theta}_{\text{MAP}}) + \frac{1}{2} \log |\boldsymbol{\Sigma}| \right)$
**end for**

---

### E.3 Linearized prediction with Woodbury

For prediction, we need to use Woodbury to compute:

$$\boldsymbol{\Sigma}_{\text{post}} = \left( (\mathbf{L} \otimes \mathbf{B})(\mathbf{L} \otimes \mathbf{B})^T + \frac{1}{\sigma^2} \mathbf{I} \otimes \mathbf{I} \right)^{-1} \tag{33}$$

$$\boldsymbol{\Sigma}_{\text{post}} = \sigma^2 \mathbf{I}_{n_{\text{lora}}} \otimes \mathbf{I}_d - \sigma^4 (\mathbf{L} \otimes \mathbf{B}) \mathbf{M}^{-1} (\mathbf{L} \otimes \mathbf{B})^T \tag{34}$$

Now, applying this form for prediction,

$$\Lambda_{ij} = \mathbf{g}_i^T \boldsymbol{\Sigma}_{\text{post}} \mathbf{g}_j. \tag{35}$$

where $\mathbf{g}_i$ is the gradient of the $i$th network output for a particular parameter, i.e. part of the Jacobian,

$$\mathbf{g}_i = \frac{\partial f_i}{\partial \boldsymbol{\theta}} \tag{36}$$

This is really a sum over layers, so we now just think of a single layer. Thus, $\mathbf{g}_i \in \mathbb{R}^{n_{\text{lora}} d}$ is really the whole matrix of gradients,

$$\mathbf{g}_i = \text{vec}(\mathbf{G}_i). \tag{37}$$

where $\mathbf{G}_i \in \mathbb{R}^{d \times n_{\text{lora}}}$. Thus,

$$\Lambda_{ij} = \text{vec}(\mathbf{G}_i)^T \boldsymbol{\Sigma}_{\text{post}} \text{vec}(\mathbf{G}_j) \tag{38}$$

$$\Lambda_{ij} = \text{vec}(\mathbf{G}_i)^T \left( \sigma^2 \mathbf{I}_{n_{\text{lora}}} \otimes \mathbf{I}_d - \sigma^4 (\mathbf{L} \otimes \mathbf{B}) \mathbf{M}^{-1} (\mathbf{L} \otimes \mathbf{B})^T \right) \text{vec}(\mathbf{G}_j) \tag{39}$$

$$\Lambda_{ij} = \sigma^2 \text{vec}(\mathbf{G}_i)^T \text{vec}(\mathbf{G}_j) - \sigma^4 \text{vec}(\mathbf{G}_i)^T (\mathbf{L} \otimes \mathbf{B}) \mathbf{M}^{-1} (\mathbf{L}^T \otimes \mathbf{B}^T) \text{vec}(\mathbf{G}_j) \tag{40}$$

Using the mixed Kronecker matrix-vector product,

$$\Lambda_{ij} = \sigma^2 \text{vec}(\mathbf{G}_i)^T \text{vec}(\mathbf{G}_j) - \sigma^4 \text{vec}(\mathbf{B}^T \mathbf{G}_i \mathbf{L})^T \mathbf{M}^{-1} \text{vec}(\mathbf{B}^T \mathbf{G}_j \mathbf{L}). \tag{41}$$

## F Optimizing Laplace prior precision

In this section, we present how we optimize the Laplace prior precision. When there is only a training set available with no validation set (such as in Figure 1, 3, 4 and Table 1, 9, 10), we can use the Laplace model evidence in Equation 14 to optimize prior precision. Our algorithm is presented in Algorithm 2, and we chose $\eta = 0.1$, and $M = 100$.

When we introduce a validation set by splitting the training set (such as in Figure 7, 5, 6 and Table 2, 11, 12), we can use the validation log-likelihood to optimize the Laplace prior precision. For memory and computational efficiency, we precompute the mean $f_{\boldsymbol{\theta}_{\text{MAP}}}$ and the Jacobian $\mathbf{J} = \nabla_{\boldsymbol{\theta}} f_{\boldsymbol{\theta}}(\mathbf{X})|_{\boldsymbol{\theta}_{\text{MAP}}}$, then perform mini-batch gradient descent on $\lambda$ (reparametrization in Bayesian model average allows gradient flowing through) as detailed in Algorithm 3. We chose $\eta = 0.1$, $M = 1000$, and $b = 4$.

## G Additional experiments

### G.1 LLaMA2-7B evaluations under distribution shift

Table 8 shows the LlaMA2-7B evaluations under distribution shift results using the best validation accuracy checkpoint on OBQA.

---

**Algorithm 3** Optimize Laplace prior precision using validation log-likelihood

---

Initialize prior precision $\lambda$, learning rate $\eta$, batch size $b$, validation set $(\mathbf{X}, \mathbf{y})$
Obtain $\boldsymbol{\theta}_{\text{MAP}}$ from a fine-tuned checkpoint, pre-compute mean $f_{\boldsymbol{\theta}_{\text{MAP}}}$, Jacobian $\mathbf{J} = \nabla_{\boldsymbol{\theta}} f_{\boldsymbol{\theta}}(\mathbf{X})|_{\boldsymbol{\theta}_{\text{MAP}}}$,
Fisher $\mathbf{F}$
**for** $step = 1, ..., M$ **do**
    Randomly sample a batch of validation data $\mathbf{X}_b, \mathbf{y}_b$ with corresponding Jacobian $\mathbf{J}_b$
    Compute posterior covariance $\boldsymbol{\Sigma} = \mathbf{F} + \lambda\mathbf{I}$
    Calculate batch logits covariance $\boldsymbol{\Lambda}_b = \mathbf{J}_b^T \boldsymbol{\Sigma} \mathbf{J}_b$ and Cholesky $\boldsymbol{\Lambda}_b = \mathbf{L}_b \mathbf{L}_b^T$
    Obtain batch Bayesian model average $\tilde{f}_{\boldsymbol{\theta}}(\mathbf{X}_b) = f_{\boldsymbol{\theta}_{\text{MAP}}}(\mathbf{X}_b) + \mathbf{L}_b\boldsymbol{\xi}$
    Evaluate validation likelihood $\text{P}\left(\mathbf{y}_b|\mathbf{X}_b, \boldsymbol{\theta}\right) = \text{Categorical}\left(\mathbf{y}_b; \text{softmax}(\tilde{f}_{\boldsymbol{\theta}}(\mathbf{X}_b))\right)$
    Perform a gradient step with respect to $\lambda$ to maximize mini-batch validation log-likelihood:
        $\lambda \leftarrow \lambda + \eta\nabla_{\lambda}\log\text{P}\left(\mathbf{y}_b|\mathbf{X}_b, \boldsymbol{\theta}\right)$
**end for**

---

Table 8: Comparison of different post-hoc methods across six out-of-distribution datasets. Results are evaluated using the fine-tuned LlaMA2-7B on OBQA as in Table 2.

| Metrics | Methods | ID | Smaller Distribution Shift | | Larger Distribution Shift | | | |
| --- | --- | --- | --- | --- | --- | --- | --- | --- |
| | | OBQA | ARC-C | ARC-E | CS | Eng | Law | Health |
| ACC ↑ | MAP | $77.7_{0.8}$ | $68.0_{0.2}$ | $76.7_{1.0}$ | $43.5_{0.9}$ | $44.4_{2.0}$ | $37.4_{0.1}$ | $47.7_{0.8}$ |
| | MC Drop | $77.7_{0.2}$ | $68.7_{0.2}$ | $77.2_{1.2}$ | $42.2_{1.3}$ | $45.6_{1.6}$ | $37.2_{0.2}$ | $47.9_{0.7}$ |
| | Ckpt Ens | $78.2_{0.2}$ | $67.9_{0.2}$ | $77.8_{0.4}$ | $42.4_{0.7}$ | $43.5_{0.9}$ | $37.1_{0.4}$ | $48.1_{0.2}$ |
| | Temp | $77.7_{0.8}$ | $68.0_{0.2}$ | $76.7_{1.0}$ | $43.5_{0.9}$ | $44.4_{2.0}$ | $37.4_{0.1}$ | $47.7_{0.8}$ |
| | LLLA | $77.6_{0.7}$ | $67.8_{0.0}$ | $77.4_{0.0}$ | $42.9_{0.0}$ | $43.8_{0.0}$ | $36.9_{0.0}$ | $46.6_{0.0}$ |
| | LA | $78.1_{0.7}$ | $67.8_{0.0}$ | $76.7_{0.0}$ | $44.1_{0.0}$ | $45.8_{0.0}$ | $37.2_{0.0}$ | $46.6_{0.0}$ |
| ECE ↓ | MAP | $9.8_{1.0}$ | $12.2_{0.7}$ | $8.6_{1.5}$ | $22.0_{1.5}$ | $20.7_{1.0}$ | $28.4_{0.8}$ | $23.2_{0.7}$ |
| | MC Drop | $8.8_{0.8}$ | $12.9_{0.8}$ | $7.7_{1.8}$ | $22.8_{2.0}$ | $19.6_{1.5}$ | $28.5_{0.6}$ | $22.6_{0.6}$ |
| | Ckpt Ens | $4.7_{0.5}$ | $7.1_{1.0}$ | $3.5_{1.1}$ | $17.4_{0.4}$ | $15.9_{1.1}$ | $25.8_{0.4}$ | $17.5_{0.4}$ |
| | Temp | $7.2_{2.6}$ | $9.4_{3.5}$ | $6.3_{1.3}$ | $16.4_{5.5}$ | $16.1_{4.6}$ | $22.7_{5.8}$ | $17.4_{6.0}$ |
| | LLLA | $5.4_{0.4}$ | $6.5_{0.0}$ | $4.9_{0.0}$ | $\mathbf{12.6_{0.0}}$ | $\mathbf{12.6_{0.0}}$ | $\mathbf{15.7_{0.0}}$ | $\mathbf{15.3_{0.0}}$ |
| | LA | $\mathbf{3.5_{0.4}}$ | $\mathbf{5.5_{0.0}}$ | $\mathbf{3.1_{0.0}}$ | $14.5_{0.0}$ | $12.8_{0.0}$ | $23.9_{0.0}$ | $17.6_{0.0}$ |
| NLL ↓ | MAP | $0.71_{0.03}$ | $0.95_{0.02}$ | $0.69_{0.02}$ | $1.40_{0.05}$ | $1.42_{0.07}$ | $1.81_{0.01}$ | $1.48_{0.02}$ |
| | MC Drop | $0.71_{0.04}$ | $0.94_{0.02}$ | $0.68_{0.03}$ | $1.40_{0.05}$ | $1.41_{0.06}$ | $1.80_{0.01}$ | $1.47_{0.02}$ |
| | Ckpt Ens | $0.65_{0.02}$ | $0.87_{0.00}$ | $\mathbf{0.61_{0.01}}$ | $1.29_{0.01}$ | $1.29_{0.03}$ | $1.71_{0.00}$ | $1.34_{0.01}$ |
| | Temp | $0.67_{0.02}$ | $0.90_{0.05}$ | $0.66_{0.01}$ | $1.31_{0.06}$ | $1.32_{0.07}$ | $1.65_{0.16}$ | $1.36_{0.10}$ |
| | LLLA | $0.66_{0.02}$ | $0.88_{0.00}$ | $0.64_{0.00}$ | $1.28_{0.00}$ | $\mathbf{1.27_{0.00}}$ | $\mathbf{1.49_{0.00}}$ | $\mathbf{1.31_{0.00}}$ |
| | LA | $\mathbf{0.62_{0.01}}$ | $\mathbf{0.85_{0.00}}$ | $0.62_{0.00}$ | $\mathbf{1.26_{0.00}}$ | $\mathbf{1.27_{0.00}}$ | $1.68_{0.00}$ | $1.35_{0.00}$ |

## G.2 FINE-TUNING ROBERTA FOR TEXT CLASSIFICATION

In this section, we present additional results of fine-tuning RoBERTa-base (Fig. 3) and RoBERTa-large (Fig. 4) (Liu et al., 2019) with LoRA on text classification tasks from GLUE (Wang et al., 2019b) and SuperGLUE (Wang et al., 2019a). Results for RoBERTa-base are shown in Figure 3 and Table 9, and results for RoBERTa-large are shown in Figure 4 and Table 10. Surprisingly, checkpoint ensemble and Monte-Carlo (MC) dropout exhibit distinct behavior on RoBERTa models compared to LlaMA2-7B. Checkpoint ensemble often performs much worse than the Maximum a Posteriori (MAP) estimation in terms of Expected Calibration Error (ECE) and Negative Log-Likelihood (NLL), while MC dropout often offers much more improvement on RoBERTa models. We suspect this difference arises due to an extra hidden penultimate layer with an additional dropout layer in front in the default RoBERTa fine-tuning setup; whereas, in LlaMA2-7B fine-tuning, we have LoRA weights and dropout on LoRA layers at the end. However, the gain from MC dropout diminishes on RoBERTa-large compared to RoBERTa-base. On the other hand, Laplace-LoRA (LA) consistently delivers substantial gains on any models we have tested (RoBERTa-base, RoBERTa-large, and LlaMA2-7B), demonstrating the robustness of Laplace-LoRA. Moreover, the Last-Layer Laplace-LoRA (LLLA) offers modest improvements as in LlaMA2-7B when optimized by Laplace model evidence, underscoring the significance of performing Bayesian inference on all LoRA weights.

Similarly, we also conduct experiments by splitting the training set into a 80% training set and a 20% validation set, then tune temperature and Laplace prior precision on the validation set. The results are

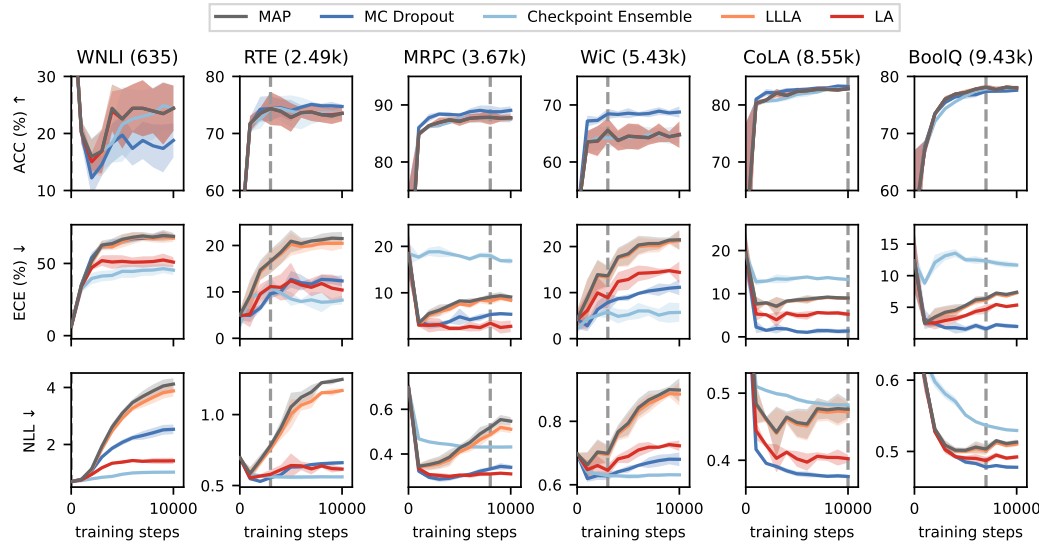

Figure 3: Fine-tuning of RoBERTa-base across six GLUE and SuperGLUE tasks (presented column-wise, with number of training examples in brackets), evaluated on the test set every 1000 gradient steps, without a validation set for hyperparameter tuning. The vertical dashed line gives the number of training steps with optimal MAP performance. Note that RoBERTa-base seems to fail on WNLI, but RoBERTa-large succeeds (Fig. 4).

Table 9: Comparison of different post-hoc methods applied to the fine-tuned RoBERTa-base across six GLUE and SuperGLUE tasks, without a validation set for hyperparameter tuning. Results are evaluated at the early stopping point of 5000 gradient steps.

| Methods | Metrics | WNLI | RTE | MRPC | WiC | CoLA | BoolQ |
|---|---|---|---|---|---|---|---|
| ACC ↑ | MAP | $22.5_{4.6}$ | $72.8_{2.2}$ | $87.1_{0.7}$ | $64.5_{2.0}$ | $82.4_{0.6}$ | $77.2_{0.4}$ |
| | MC Drop | $19.7_{3.0}$ | $74.0_{1.6}$ | $88.2_{0.0}$ | $68.5_{0.8}$ | $82.7_{0.1}$ | $76.7_{0.6}$ |
| | Ckpt Ens | $21.6_{5.8}$ | $73.9_{2.5}$ | $87.7_{0.6}$ | $64.3_{1.9}$ | $81.8_{0.5}$ | $76.6_{0.2}$ |
| | LLLA | $22.5_{4.6}$ | $72.8_{2.2}$ | $87.1_{0.7}$ | $64.5_{2.0}$ | $82.4_{0.6}$ | $77.2_{0.4}$ |
| | LA | $22.5_{4.6}$ | $72.8_{2.2}$ | $87.2_{0.7}$ | $64.5_{2.0}$ | $82.4_{0.6}$ | $77.2_{0.5}$ |
| ECE ↓ | MAP | $66.7_{3.2}$ | $20.9_{2.1}$ | $8.3_{0.1}$ | $18.4_{2.2}$ | $8.6_{0.5}$ | $5.3_{0.2}$ |
| | MC Drop | $65.6_{1.8}$ | $12.7_{1.6}$ | $4.8_{1.6}$ | $8.9_{0.6}$ | $\mathbf{1.2_{0.1}}$ | $\mathbf{1.5_{0.4}}$ |
| | Ckpt Ens | $\mathbf{44.6_{3.9}}$ | $\mathbf{8.5_{1.3}}$ | $17.9_{0.1}$ | $\mathbf{5.2_{1.0}}$ | $13.6_{0.3}$ | $12.5_{0.3}$ |
| | LLLA | $65.2_{3.7}$ | $20.3_{2.2}$ | $7.6_{0.3}$ | $18.1_{2.2}$ | $8.5_{0.4}$ | $5.1_{0.1}$ |
| | LA | $51.4_{3.8}$ | $12.4_{3.4}$ | $\mathbf{2.3_{0.4}}$ | $12.9_{2.0}$ | $5.0_{0.6}$ | $3.7_{0.2}$ |
| NLL ↓ | MAP | $3.10_{0.09}$ | $1.05_{0.09}$ | $0.43_{0.01}$ | $0.79_{0.03}$ | $0.46_{0.02}$ | $0.50_{0.00}$ |
| | MC Drop | $2.07_{0.08}$ | $0.63_{0.01}$ | $\mathbf{0.30_{0.01}}$ | $0.65_{0.01}$ | $\mathbf{0.38_{0.00}}$ | $\mathbf{0.49_{0.00}}$ |
| | Ckpt Ens | $\mathbf{0.96_{0.02}}$ | $\mathbf{0.56_{0.01}}$ | $0.44_{0.00}$ | $\mathbf{0.63_{0.01}}$ | $0.49_{0.00}$ | $0.55_{0.00}$ |
| | LLLA | $2.92_{0.08}$ | $1.00_{0.09}$ | $0.41_{0.01}$ | $0.79_{0.03}$ | $0.45_{0.02}$ | $0.50_{0.00}$ |
| | LA | $1.38_{0.06}$ | $0.64_{0.08}$ | $\mathbf{0.30_{0.00}}$ | $0.69_{0.02}$ | $0.40_{0.01}$ | $\mathbf{0.49_{0.00}}$ |

shown in Figure 5 and Table 11 for RoBERTa-base, and Figure 6 and Table 12 for RoBERTa-large. Again, full Laplace-LoRA LA offers the most improvements most of the time, and is the most robust method overall.

## G.3 FINE-TUNING LLAMA2-7B FOR COMMON SENSE REASONING

Figure 7 displays the results of tuning temperature scaling and Laplace prior precision on a held-out validation set split from the training set. LLLA is missing a few zero checkpoint evaluation results due to Cholesky errors. Comparing Figure 7 to Figure 1 in the main text, it is evident that splitting the training set into a smaller training set (80% data) and a validation set (20% data) slightly impact the accuracy of fine-tuned model.

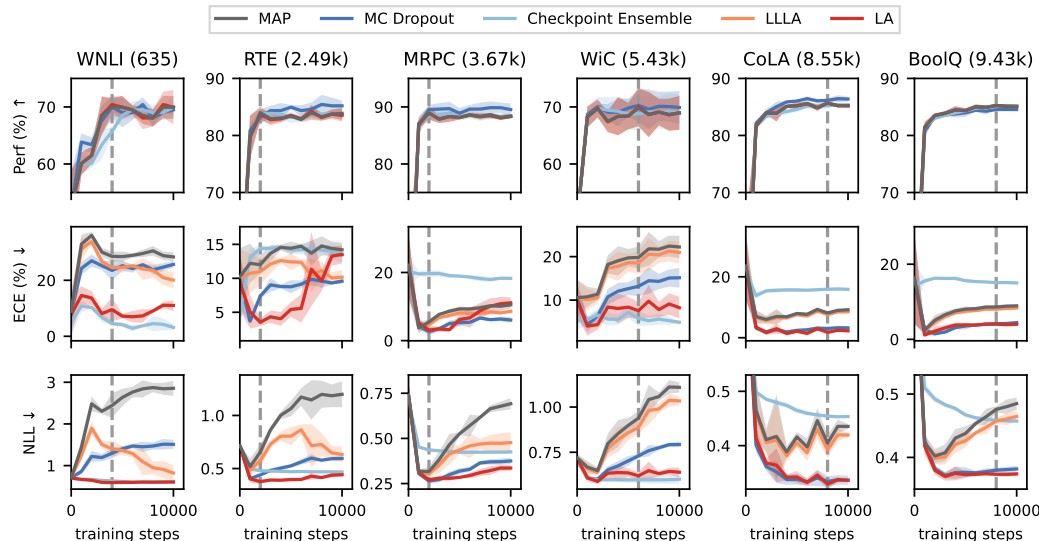

Figure 4: Fine-tuning RoBERTa-large across the six GLUE and SuperGLUE tasks in Fig. 3, without a validation set for hyperparameter tuning. The vertical dashed line gives the number of training steps with optimal MAP performance.

Table 10: Comparison of different post-hoc methods applied to the fine-tuned RoBERTa-large across six GLUE and SuperGLUE tasks. Results are evaluated at the early stopping point of 5000 gradient steps.

| Methods | Metrics | WNLI | RTE | MRPC | WiC | CoLA | BoolQ |
|---|---|---|---|---|---|---|---|
| ACC ↑ | MAP | $70.0_{1.8}$ | $83.4_{0.3}$ | $88.2_{0.3}$ | $68.4_{2.0}$ | $85.2_{0.5}$ | $84.4_{0.4}$ |
| | MC Drop | $69.0_{2.3}$ | $85.0_{0.3}$ | $88.9_{0.8}$ | $69.8_{1.4}$ | $86.0_{0.1}$ | $84.0_{0.6}$ |
| | Ckpt Ens | $68.5_{0.7}$ | $83.2_{0.9}$ | $88.2_{0.5}$ | $68.8_{2.4}$ | $84.6_{0.4}$ | $84.3_{0.5}$ |
| | LLLA | $70.0_{1.8}$ | $83.5_{0.3}$ | $88.2_{0.3}$ | $68.4_{2.0}$ | $85.2_{0.5}$ | $84.3_{0.3}$ |
| | LA | $70.0_{1.8}$ | $83.5_{0.3}$ | $88.2_{0.3}$ | $68.4_{2.0}$ | $85.2_{0.5}$ | $84.4_{0.4}$ |
| ECE ↓ | MAP | $28.5_{1.6}$ | $14.4_{0.8}$ | $9.3_{0.6}$ | $19.7_{1.8}$ | $8.0_{0.4}$ | $8.0_{0.6}$ |
| | MC Drop | $25.2_{1.6}$ | $8.8_{0.9}$ | $\mathbf{5.3_{1.0}}$ | $12.6_{1.3}$ | $2.4_{0.3}$ | $\mathbf{3.6_{0.9}}$ |
| | Ckpt Ens | $\mathbf{4.2_{0.7}}$ | $14.2_{0.5}$ | $18.9_{0.2}$ | $\mathbf{7.2_{1.2}}$ | $15.7_{0.5}$ | $15.6_{0.3}$ |
| | LLLA | $24.6_{1.7}$ | $12.4_{1.1}$ | $8.1_{0.6}$ | $18.6_{1.8}$ | $7.6_{0.3}$ | $7.5_{0.7}$ |
| | LA | $7.2_{0.7}$ | $\mathbf{5.4_{1.0}}$ | $6.2_{1.5}$ | $8.4_{2.1}$ | $\mathbf{2.1_{0.3}}$ | $3.8_{0.5}$ |
| NLL ↓ | MAP | $2.64_{0.18}$ | $1.06_{0.06}$ | $0.53_{0.01}$ | $0.90_{0.04}$ | $0.42_{0.01}$ | $0.44_{0.01}$ |
| | MC Drop | $1.40_{0.06}$ | $0.53_{0.01}$ | $0.32_{0.02}$ | $0.70_{0.02}$ | $\mathbf{0.34_{0.01}}$ | $0.38_{0.01}$ |
| | Ckpt Ens | $0.62_{0.01}$ | $0.47_{0.00}$ | $0.42_{0.00}$ | $\mathbf{0.60_{0.01}}$ | $0.46_{0.00}$ | $0.47_{0.00}$ |
| | LLLA | $1.36_{0.10}$ | $0.80_{0.08}$ | $0.44_{0.02}$ | $0.86_{0.04}$ | $0.41_{0.01}$ | $0.43_{0.01}$ |
| | LA | $\mathbf{0.60_{0.02}}$ | $\mathbf{0.40_{0.01}}$ | $\mathbf{0.28_{0.00}}$ | $0.64_{0.02}$ | $\mathbf{0.34_{0.01}}$ | $\mathbf{0.37_{0.00}}$ |

### G.4 Fine-tuning Mistral-7B for common sense reasoning

In this section, we present the results using different decoder-based LLM backbones, specifically Mistral-7B (Jiang et al., 2023). Table 13 shows the results of fine-tuning Mistral-7B on common-sense reasoning tasks. LA consistently offers the best NLL across all datasets, as well as 3 out of 6 best ECE, indicating the robustness of Laplace-LoRA on different decoder-only LLM backbones.

### G.5 Laplace-LoRA on different subsets of layers

Besides applying Laplace approximation on all layers or only the last layer, we can also apply Laplace approximation on a subset of layers, for example, the first $k$ layers. In Table 14, we present the results of applying Laplace approximation to the top 8 layers, 16 layers, 24 layers, and all layers. Although

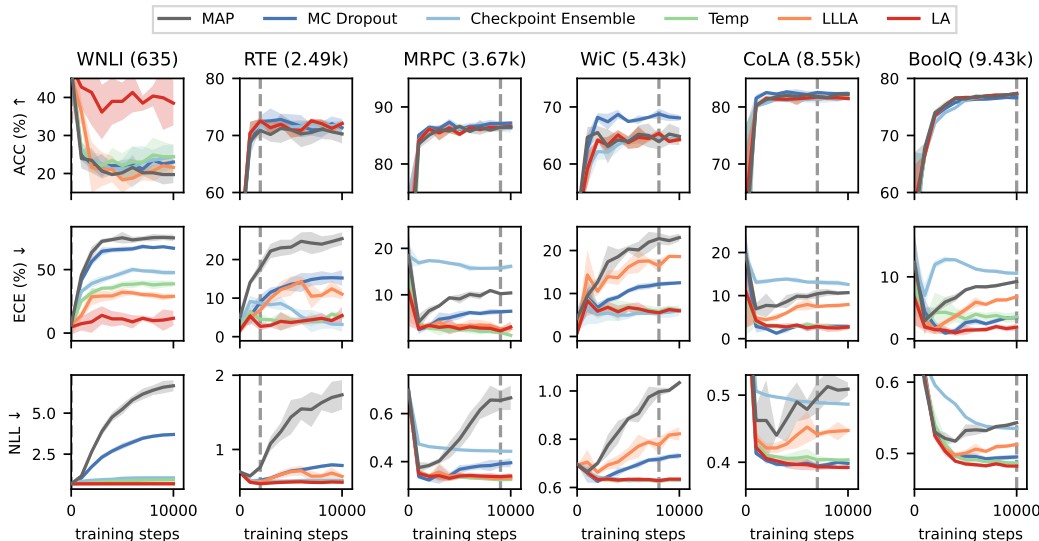

Figure 5: Fine-tuning RoBERTa-base across the six GLUE and SuperGLUE tasks, with a validation set for hyperparameter tuning. The vertical dashed line gives the checkpoint with optimal MAP performance on the validation set.

Table 11: Comparison of different post-hoc methods applied to the fine-tuned RoBERTa-base across six GLUE and SuperGLUE tasks, with a validation set for hyperparameter tuning. Results are evaluated at the best MAP performance checkpoint observed on the validation set.

| Methods | Metrics | WNLI | RTE | MRPC | WiC | CoLA | BoolQ |
|---------|---------|------|-----|------|-----|------|-------|
| ACC ↑ | MAP | $47.9_{6.0}$ | $70.9_{0.9}$ | $86.4_{0.6}$ | $63.9_{1.2}$ | $81.8_{0.3}$ | $77.2_{0.2}$ |
| | MC Drop | $47.9_{6.0}$ | $72.4_{0.9}$ | $87.1_{0.3}$ | $68.8_{0.9}$ | $82.6_{0.1}$ | $76.6_{0.4}$ |
| | Ckpt Ens | $47.9_{6.0}$ | $71.8_{0.9}$ | $86.3_{0.5}$ | $64.7_{0.3}$ | $81.4_{0.7}$ | $77.2_{0.1}$ |
| | Temp | $47.9_{6.0}$ | $72.6_{0.3}$ | $86.5_{0.3}$ | $65.4_{0.8}$ | $81.8_{0.8}$ | $77.3_{0.2}$ |
| | LLLA | $46.0_{10.0}$ | $72.6_{0.3}$ | $86.4_{0.4}$ | $65.3_{0.7}$ | $81.8_{0.7}$ | $77.4_{0.2}$ |
| | LA | $48.4_{7.7}$ | $72.6_{0.3}$ | $86.4_{0.3}$ | $65.4_{0.6}$ | $81.7_{0.7}$ | $77.4_{0.2}$ |
| ECE ↓ | MAP | $6.3_{4.3}$ | $17.8_{1.4}$ | $10.2_{0.2}$ | $22.8_{1.1}$ | $10.4_{0.9}$ | $9.2_{0.1}$ |
| | MC Drop | $6.4_{4.4}$ | $9.0_{1.7}$ | $6.3_{0.4}$ | $12.2_{0.5}$ | $\mathbf{2.8_{0.3}}$ | $3.4_{0.1}$ |
| | Ckpt Ens | $6.2_{2.3}$ | $8.8_{0.9}$ | $15.7_{0.6}$ | $5.5_{1.0}$ | $13.1_{0.5}$ | $10.6_{0.1}$ |
| | Temp | $5.5_{1.7}$ | $4.5_{1.3}$ | $\mathbf{1.9_{0.3}}$ | $5.8_{0.9}$ | $2.9_{0.6}$ | $3.5_{0.7}$ |
| | LLLA | $6.8_{7.8}$ | $6.9_{0.7}$ | $2.6_{0.4}$ | $16.4_{1.4}$ | $7.4_{0.3}$ | $6.8_{0.2}$ |
| | LA | $\mathbf{4.7_{5.2}}$ | $\mathbf{2.7_{1.0}}$ | $2.2_{0.7}$ | $\mathbf{5.5_{0.5}}$ | $\mathbf{2.8_{0.6}}$ | $\mathbf{1.9_{0.5}}$ |
| NLL ↓ | MAP | $0.70_{0.01}$ | $0.76_{0.03}$ | $0.66_{0.04}$ | $1.00_{0.02}$ | $0.50_{0.02}$ | $0.54_{0.00}$ |
| | MC Drop | $0.70_{0.01}$ | $0.58_{0.03}$ | $0.39_{0.01}$ | $0.72_{0.01}$ | $0.39_{0.00}$ | $0.50_{0.00}$ |
| | Ckpt Ens | $\mathbf{0.69_{0.00}}$ | $0.57_{0.00}$ | $0.44_{0.00}$ | $0.63_{0.00}$ | $0.49_{0.00}$ | $0.53_{0.00}$ |
| | Temp | $\mathbf{0.69_{0.00}}$ | $\mathbf{0.54_{0.00}}$ | $\mathbf{0.32_{0.00}}$ | $0.62_{0.01}$ | $0.40_{0.01}$ | $0.49_{0.00}$ |
| | LLLA | $\mathbf{0.69_{0.00}}$ | $0.56_{0.01}$ | $0.33_{0.00}$ | $0.78_{0.03}$ | $0.44_{0.00}$ | $0.51_{0.00}$ |
| | LA | $\mathbf{0.69_{0.00}}$ | $\mathbf{0.54_{0.00}}$ | $0.34_{0.01}$ | $\mathbf{0.62_{0.01}}$ | $\mathbf{0.39_{0.00}}$ | $\mathbf{0.48_{0.00}}$ |

there is no conclusive trend in the results, applying Laplace approximation to more than 8 layers usually leads to improved NLL.

## G.6 DIAGONAL LAPLACE APPROXIMATION

In this section, we present the results for Laplace-LoRA utilizing a diagonal approximation of the Hessian. This approach is generally not found to be as effective as the Kronecker-factored Approximate Curvature (K-FAC) (Daxberger et al., 2021a) that approximates the block diagonal Hessian.

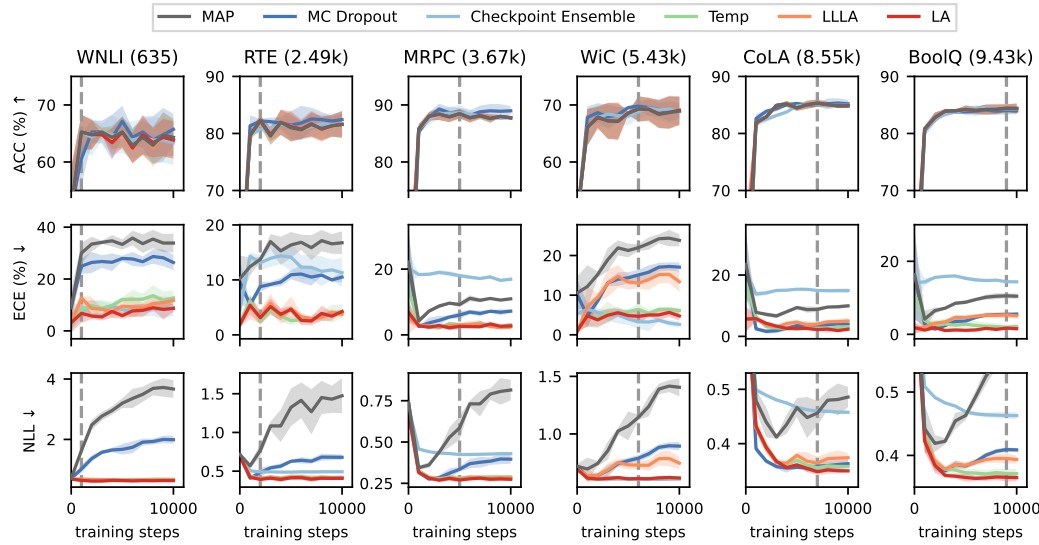

Figure 6: Fine-tuning RoBERTa-large across the six GLUE and SuperGLUE tasks, with a validation set for hyperparameter tuning. The vertical dashed line gives the checkpoint with optimal MAP performance on the validation set.

Table 12: Comparison of different post-hoc methods applied to the fine-tuned RoBERTa-large across six GLUE and SuperGLUE tasks, with a validation set split from the training set used for tuning temperature and Laplace prior precision. Results are evaluated at the best MAP performance checkpoint observed on the validation set.

| Methods | Metrics | WNLI | RTE | MRPC | WiC | CoLA | BoolQ |
|---|---|---|---|---|---|---|---|
| ACC ↑ | MAP | $65.3_{2.7}$ | $82.2_{0.9}$ | $88.5_{0.7}$ | $69.3_{1.9}$ | $85.3_{0.6}$ | $84.4_{0.6}$ |
| | MC Drop | $60.6_{4.0}$ | $82.2_{0.3}$ | $88.7_{0.7}$ | $69.8_{0.8}$ | $85.1_{0.3}$ | $84.0_{0.6}$ |
| | Ckpt Ens | $65.3_{2.7}$ | $80.7_{0.3}$ | $88.6_{1.1}$ | $68.8_{1.0}$ | $85.2_{0.4}$ | $84.3_{0.8}$ |
| | Temp | $65.3_{2.7}$ | $82.2_{0.9}$ | $88.5_{0.7}$ | $69.3_{1.9}$ | $85.3_{0.6}$ | $84.4_{0.6}$ |
| | LLLA | $65.3_{2.7}$ | $82.2_{0.9}$ | $88.5_{0.7}$ | $69.3_{1.9}$ | $85.3_{0.6}$ | $84.4_{0.6}$ |
| | LA | $65.3_{2.7}$ | $82.3_{0.8}$ | $88.5_{0.7}$ | $69.3_{2.0}$ | $85.3_{0.5}$ | $84.4_{0.6}$ |
| ECE ↓ | MAP | $30.0_{3.5}$ | $13.7_{0.4}$ | $9.2_{0.7}$ | $22.0_{0.8}$ | $8.8_{0.8}$ | $10.5_{0.6}$ |
| | MC Drop | $24.9_{2.6}$ | $8.8_{0.8}$ | $6.0_{0.9}$ | $15.1_{1.1}$ | $3.3_{0.8}$ | $5.5_{0.4}$ |
| | Ckpt Ens | $7.7_{3.0}$ | $13.1_{1.2}$ | $18.0_{0.5}$ | $\mathbf{3.3_{1.6}}$ | $15.0_{0.3}$ | $14.4_{0.1}$ |
| | Temp | $8.1_{2.2}$ | $\mathbf{2.6_{0.7}}$ | $3.1_{0.4}$ | $6.4_{1.3}$ | $2.7_{0.8}$ | $1.9_{0.7}$ |
| | LLLA | $12.7_{0.9}$ | $3.3_{1.0}$ | $3.0_{0.5}$ | $13.1_{1.2}$ | $4.0_{0.4}$ | $5.3_{0.8}$ |
| | LA | $\mathbf{6.8_{3.3}}$ | $3.1_{1.1}$ | $\mathbf{2.6_{0.7}}$ | $4.7_{1.7}$ | $\mathbf{2.2_{0.2}}$ | $\mathbf{1.7_{0.4}}$ |
| NLL ↓ | MAP | $1.63_{0.09}$ | $0.77_{0.07}$ | $0.59_{0.06}$ | $1.14_{0.02}$ | $0.46_{0.02}$ | $0.56_{0.01}$ |
| | MC Drop | $1.02_{0.13}$ | $0.48_{0.06}$ | $0.33_{0.01}$ | $0.79_{0.02}$ | $0.36_{0.01}$ | $0.41_{0.00}$ |
| | Ckpt Ens | $0.65_{0.02}$ | $0.49_{0.01}$ | $0.42_{0.00}$ | $\mathbf{0.61_{0.01}}$ | $0.46_{0.00}$ | $0.46_{0.00}$ |
| | Temp | $0.65_{0.01}$ | $0.41_{0.01}$ | $0.28_{0.02}$ | $\mathbf{0.61_{0.01}}s$ | $0.36_{0.00}$ | $0.37_{0.01}$ |
| | LLLA | $0.69_{0.02}$ | $0.40_{0.02}$ | $\mathbf{0.27_{0.01}}$ | $0.73_{0.03}$ | $0.36_{0.01}$ | $0.40_{0.01}$ |
| | LA | $\mathbf{0.64_{0.00}}$ | $\mathbf{0.39_{0.01}}$ | $\mathbf{0.27_{0.01}}$ | $\mathbf{0.61_{0.01}}$ | $\mathbf{0.35_{0.01}}$ | $\mathbf{0.37_{0.01}}$ |

### G.6.1 ROBERTA

The results on RoBERTa-base and RoBERTa-large are shown in Figure 8 Table 15, and Figure 9 Table 16. LLLA still offers a tiny advantage over Maximum a Posteriori (MAP) in ECE and NLL, however, LA show mixed performance across the datasets.

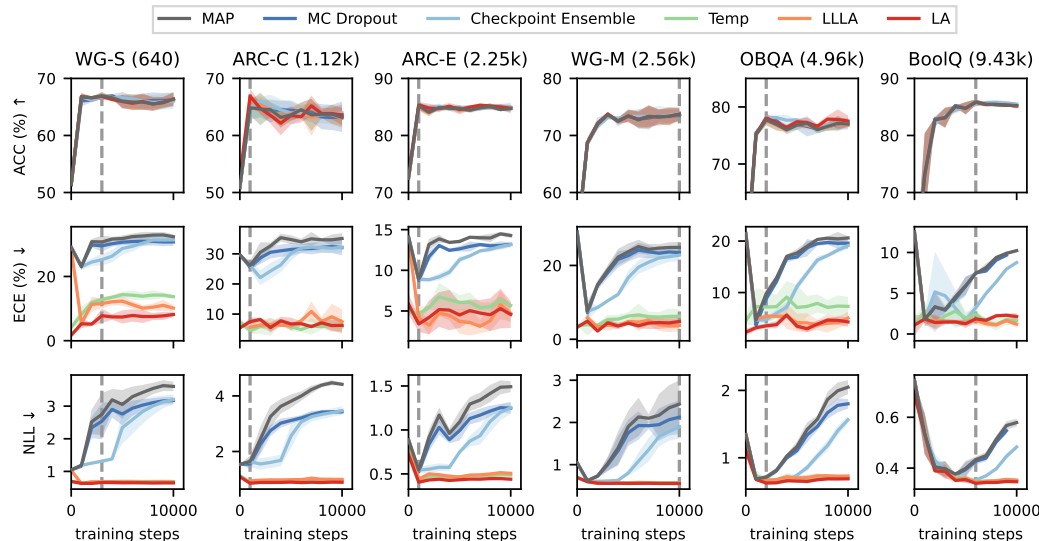

Figure 7: Fine-tuning of LlaMA2-7B across six common sense reasoning tasks (presented column-wise, with number of training examples in brackets), evaluated on the test set every 1000 gradient steps. The vertical dashed line gives the checkpoint with optimal MAP performance on a held-out validation set.

Table 13: Comparison of different post-hoc methods applied to the fine-tuned Mistral-7B across six common sense reasoning tasks. Results are evaluated at the early stopping point of 5000 gradient steps. We report standard deviations in subscripts, and bold numbers that are statistical significant.

| Methods | Metrics | WG-S | ARC-C | ARC-E | WG-M | OBQA | BoolQ |
|---|---|---|---|---|---|---|---|
| ACC ↑ | MAP | $77.4_{0.5}$ | $82.3_{1.0}$ | $90.7_{0.4}$ | $83.1_{0.8}$ | $87.0_{0.5}$ | $89.2_{0.1}$ |
|  | MC Drop | $77.4_{0.4}$ | $81.6_{1.4}$ | $90.7_{0.5}$ | $83.0_{1.0}$ | $87.1_{0.8}$ | $89.3_{0.1}$ |
|  | Ckpt Ens | $77.2_{0.2}$ | $81.8_{1.7}$ | $90.7_{0.7}$ | $83.8_{0.5}$ | $88.3_{0.5}$ | $89.0_{0.3}$ |
|  | LLLA | $77.4_{0.5}$ | $81.7_{1.0}$ | $90.6_{0.4}$ | $83.1_{0.8}$ | $87.0_{0.5}$ | $89.2_{0.1}$ |
|  | LA | $77.2_{0.3}$ | $79.9_{3.3}$ | $89.4_{0.9}$ | $83.0_{0.8}$ | $87.6_{0.8}$ | $89.2_{0.1}$ |
| ECE ↓ | MAP | $22.0_{0.6}$ | $16.1_{1.3}$ | $8.5_{0.6}$ | $13.5_{1.5}$ | $9.3_{0.0}$ | $5.6_{0.6}$ |
|  | MC Drop | $20.5_{0.3}$ | $15.2_{1.5}$ | $7.6_{0.4}$ | $13.0_{1.6}$ | $8.8_{0.5}$ | $5.3_{0.6}$ |
|  | Ckpt Ens | $21.1_{0.3}$ | $\mathbf{14.7_{2.3}}$ | $\mathbf{7.1_{0.8}}$ | $10.1_{0.6}$ | $\mathbf{6.3_{0.3}}$ | $2.0_{0.6}$ |
|  | LLLA | $10.2_{0.5}$ | $29.4_{3.1}$ | $19.1_{7.7}$ | $12.3_{1.4}$ | $7.6_{0.4}$ | $5.4_{0.5}$ |
|  | LA | $\mathbf{8.2_{0.3}}$ | $21.8_{6.7}$ | $21.6_{8.1}$ | $\mathbf{6.6_{1.7}}$ | $7.2_{0.9}$ | $\mathbf{1.8_{0.4}}$ |
| NLL ↓ | MAP | $2.38_{0.27}$ | $1.60_{0.16}$ | $0.75_{0.12}$ | $0.80_{0.14}$ | $0.57_{0.03}$ | $0.32_{0.02}$ |
|  | MC Drop | $2.12_{0.20}$ | $1.41_{0.10}$ | $0.62_{0.07}$ | $0.75_{0.12}$ | $0.54_{0.03}$ | $0.31_{0.02}$ |
|  | Ckpt Ens | $2.04_{0.13}$ | $1.33_{0.16}$ | $\mathbf{0.48_{0.02}}$ | $0.55_{0.02}$ | $0.45_{0.01}$ | $\mathbf{0.27_{0.01}}$ |
|  | LLLA | $0.54_{0.01}$ | $0.82_{0.01}$ | $\mathbf{0.48_{0.08}}$ | $0.67_{0.07}$ | $0.47_{0.05}$ | $0.32_{0.02}$ |
|  | LA | $\mathbf{0.52_{0.02}}$ | $\mathbf{0.80_{0.15}}$ | $\mathbf{0.48_{0.11}}$ | $\mathbf{0.42_{0.01}}$ | $\mathbf{0.37_{0.01}}$ | $\mathbf{0.27_{0.01}}$ |

### G.6.2 LLAMA2-7B

On LlaMA2-7B, both LLLA and LA show mixed performance and do not offer consistent improvements as shown in Figure 10 and Table 17. Specifically, the accuracy of diagonal LA is much worse than MAP on ARC datasets, and ECE is worse than MAP on ARC-E, OBQA and BoolQ.

On the other hand, when we split a validation set from the training set and tune the Laplace prior precision using the validation log-likelihood, the results are much better, as shown in Figure 11 and Table 18.

Table 14: Laplace-LoRA applied to different subsets of layers on the fine-tuned LlaMA2-7B across six common sense reasoning tasks. Results are evaluated at the early stopping point of 5000 gradient steps. We report standard deviations in subscripts, and bold numbers that are statistical significant.

| Methods | Metrics | WG-S | ARC-C | ARC-E | WG-M | OBQA | BoolQ |
|---|---|---|---|---|---|---|---|
| ACC ↑ | LA 8 | $67.4_{0.8}$ | $65.6_{0.4}$ | $84.7_{1.5}$ | $73.3_{0.8}$ | $79.3_{0.4}$ | $85.9_{0.3}$ |
| | LA 16 | $67.6_{0.8}$ | $65.6_{0.4}$ | $84.6_{1.4}$ | $73.4_{0.7}$ | $79.3_{0.2}$ | $85.9_{0.3}$ |
| | LA 24 | $67.5_{0.8}$ | $65.1_{0.5}$ | $84.5_{1.4}$ | $73.4_{0.7}$ | $79.3_{0.2}$ | $85.9_{0.3}$ |
| | LA | $67.3_{0.2}$ | $65.3_{0.2}$ | $85.1_{1.5}$ | $73.4_{0.3}$ | $78.9_{0.2}$ | $86.1_{0.2}$ |
| ECE ↓ | LA 8 | $6.4_{0.9}$ | $3.6_{0.4}$ | $4.3_{1.6}$ | $8.7_{0.6}$ | $6.3_{0.3}$ | $2.3_{0.5}$ |
| | LA 16 | $2.8_{1.0}$ | $5.8_{0.6}$ | $5.3_{0.3}$ | $5.8_{0.5}$ | $4.6_{0.7}$ | $1.4_{0.0}$ |
| | LA 24 | $2.0_{1.1}$ | $7.2_{1.1}$ | $6.6_{1.7}$ | $5.2_{0.5}$ | $3.7_{0.4}$ | $1.3_{0.0}$ |
| | LA | $2.1_{0.3}$ | $7.4_{0.7}$ | $5.4_{0.2}$ | $7.4_{0.4}$ | $6.4_{0.8}$ | $2.1_{0.6}$ |
| NLL ↓ | LA 8 | $0.63_{0.00}$ | $0.87_{0.03}$ | $0.50_{0.08}$ | $0.67_{0.01}$ | $0.68_{0.01}$ | $0.34_{0.00}$ |
| | LA 16 | $0.61_{0.01}$ | $0.86_{0.02}$ | $0.48_{0.06}$ | $0.59_{0.01}$ | $0.63_{0.01}$ | $0.33_{0.00}$ |
| | LA 24 | $0.60_{0.01}$ | $0.87_{0.02}$ | $0.48_{0.05}$ | $0.58_{0.01}$ | $0.62_{0.01}$ | $0.33_{0.00}$ |
| | LA | $0.60_{0.01}$ | $0.88_{0.03}$ | $0.49_{0.06}$ | $0.63_{0.02}$ | $0.65_{0.01}$ | $0.34_{0.01}$ |

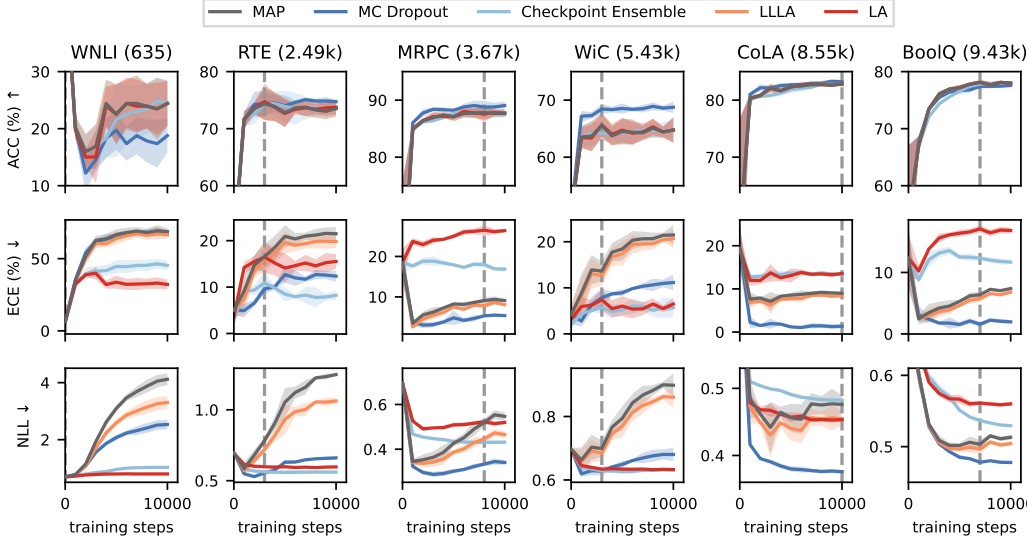

Figure 8: Fine-tuning of RoBERTa-base across six GLUE and SuperGLUE tasks (presented column-wise, with number of training examples in brackets), evaluated on the test set every 1000 gradient steps. The vertical dashed line gives the number of training steps with optimal MAP performance. LA and LLLA using diagonal Fisher approximation.

# H   LA vs LLLA, or which layers are uncertain?

When comparing LA and LLLA, it is important to understand where the uncertainty arose from: the last layer (which is the only thing that LLLA captures), or the earlier layers (which is also captured by LA). To understand this, we plotted the standard deviation of the logits arising from various sources (Fig. 12). We found that for full Laplace-LoRA LA, almost all the uncertainty arose from the lower-layers (green solid bar), rather than the last layer (blue solid bar). Indeed, when optimizing the prior precisions using the model evidence, this implied that LLLA (orange solid bar) gave rise to far less uncertainty in the logits than LA. This is likely a good explanation for the poor performance of LLLA when optimizing using the model evidence (Table 1). However, optimizing the prior precisions using the validation LL gives radically different results for LLLA, with much higher logit variances (dashed orange bar), and reasonable performance (Table 2). To understand what is going on here, consider a setting where alot of variability in the logits is necessary to optimize the validation LL. Then if we use the validation LL directly as an objective, LLLA can just increase the prior precision until the variance in the logits is high enough. While this may be a reasonable strategy, the resulting

Table 15: Comparison of different post-hoc methods applied to the fine-tuned RoBERTa-base across six common GLUE and SuperGLUE tasks. Results are evaluated at the early stopping point of 5000 gradient steps. LA and LLLA using diagonal Fisher approximation.

| Methods | Metrics | WNLI | RTE | MRPC | WiC | CoLA | BoolQ |
|---|---|---|---|---|---|---|---|
| ACC ↑ | MAP | $22.5_{4.6}$ | $72.8_{2.2}$ | $87.1_{0.7}$ | $64.5_{2.0}$ | $82.4_{0.6}$ | $77.2_{0.4}$ |
|  | MC Drop | $19.7_{3.0}$ | $74.0_{1.6}$ | $88.2_{0.0}$ | $68.5_{0.8}$ | $82.7_{0.1}$ | $76.7_{0.6}$ |
|  | Ckpt Ens | $21.6_{5.8}$ | $73.9_{2.5}$ | $87.7_{0.6}$ | $64.3_{1.9}$ | $81.8_{0.5}$ | $76.6_{0.2}$ |
|  | LLLA | $22.5_{4.6}$ | $72.8_{2.2}$ | $87.1_{0.7}$ | $64.5_{2.0}$ | $82.4_{0.6}$ | $77.2_{0.4}$ |
|  | LA | $22.5_{4.6}$ | $72.7_{2.1}$ | $87.1_{0.5}$ | $64.7_{1.9}$ | $82.3_{0.7}$ | $77.2_{0.5}$ |
| ECE ↓ | MAP | $66.7_{3.2}$ | $20.9_{2.1}$ | $8.3_{0.1}$ | $18.4_{2.2}$ | $8.6_{0.5}$ | $5.3_{0.2}$ |
|  | MC Drop | $65.6_{1.8}$ | $12.7_{1.6}$ | $\mathbf{4.8_{1.6}}$ | $8.9_{0.6}$ | $\mathbf{1.2_{0.1}}$ | $\mathbf{1.5_{0.4}}$ |
|  | Ckpt Ens | $44.6_{3.9}$ | $\mathbf{8.5_{1.3}}$ | $17.9_{0.1}$ | $\mathbf{5.2_{1.0}}$ | $13.6_{0.3}$ | $12.5_{0.3}$ |
|  | LLLA | $64.0_{3.7}$ | $19.6_{2.1}$ | $7.5_{0.2}$ | $17.4_{2.3}$ | $7.7_{0.5}$ | $4.6_{0.1}$ |
|  | LA | $\mathbf{33.7_{4.0}}$ | $14.1_{2.8}$ | $25.0_{0.5}$ | $6.0_{1.9}$ | $13.1_{1.4}$ | $16.4_{0.5}$ |
| NLL ↓ | MAP | $3.10_{0.09}$ | $1.05_{0.09}$ | $0.43_{0.01}$ | $0.79_{0.03}$ | $0.46_{0.02}$ | $0.50_{0.00}$ |
|  | MC Drop | $2.07_{0.08}$ | $0.63_{0.01}$ | $\mathbf{0.30_{0.01}}$ | $0.65_{0.01}$ | $\mathbf{0.38_{0.00}}$ | $\mathbf{0.49_{0.00}}$ |
|  | Ckpt Ens | $0.96_{0.02}$ | $\mathbf{0.56_{0.01}}$ | $0.44_{0.00}$ | $\mathbf{0.63_{0.01}}$ | $0.49_{0.00}$ | $0.55_{0.00}$ |
|  | LLLA | $2.59_{0.10}$ | $0.93_{0.09}$ | $0.39_{0.01}$ | $0.77_{0.03}$ | $0.44_{0.02}$ | $0.50_{0.00}$ |
|  | LA | $\mathbf{0.80_{0.00}}$ | $0.60_{0.01}$ | $0.51_{0.01}$ | $\mathbf{0.63_{0.00}}$ | $0.46_{0.01}$ | $0.57_{0.01}$ |

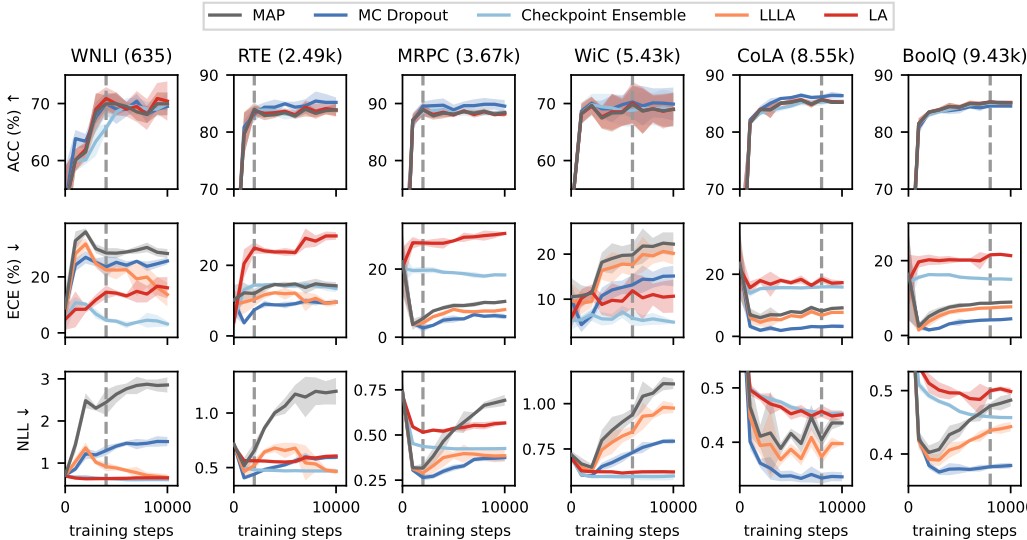

Figure 9: Fine-tuning of RoBERTa-large across six GLUE and SuperGLUE tasks (presented column-wise, with number of training examples in brackets), evaluated on the test set every 1000 gradient steps. The vertical dashed line gives the number of training steps with optimal MAP performance. LA and LLLA using diagonal Fisher approximation.

validation-LL optimized LLLA posteriors differ dramatically from those we would expect under Bayes, where the uncertainty in at the last layer weights is far lower, and most of the uncertainty in logits arises at lower layers.

# I EVALUATION DATASETS UNDER LARGER DISTRIBUTION SHIFT

Here we present the specific MMLU datasets we used for evaluations under distribution shift in Table 3 and Table 8. The specific task splits we selected for each subject are shown in Table 19 assigned by Hendrycks et al. (2020).

Table 16: Comparison of different post-hoc methods applied to the fine-tuned RoBERTa-large across six common GLUE and SuperGLUE tasks. Results are evaluated at the early stopping point of 5000 gradient steps. LA and LLLA using diagonal Fisher approximation.

| Methods | Metrics | WNLI | RTE | MRPC | WiC | CoLA | BoolQ |
|---------|---------|------|-----|------|-----|------|-------|
| ACC ↑ | MAP | $70.0_{1.8}$ | $83.4_{0.3}$ | $88.2_{0.3}$ | $68.4_{2.0}$ | $85.2_{0.5}$ | $84.4_{0.4}$ |
| | MC Drop | $69.0_{2.3}$ | $85.0_{0.3}$ | $88.9_{0.8}$ | $69.8_{1.4}$ | $86.0_{0.1}$ | $84.0_{0.6}$ |
| | Ckpt Ens | $68.5_{0.7}$ | $83.2_{0.9}$ | $88.2_{0.5}$ | $68.8_{2.4}$ | $84.6_{0.4}$ | $84.3_{0.5}$ |
| | LLLA | $70.0_{1.8}$ | $83.5_{0.3}$ | $88.2_{0.3}$ | $68.4_{2.0}$ | $85.2_{0.5}$ | $84.4_{0.4}$ |
| | LA | $70.0_{1.8}$ | $83.6_{0.5}$ | $88.0_{0.3}$ | $68.5_{2.2}$ | $85.3_{0.6}$ | $84.4_{0.4}$ |
| ECE ↓ | MAP | $28.5_{1.6}$ | $14.4_{0.8}$ | $9.3_{0.6}$ | $19.7_{1.8}$ | $8.0_{0.4}$ | $8.0_{0.6}$ |
| | MC Drop | $25.2_{1.6}$ | $\mathbf{8.8_{0.9}}$ | $\mathbf{5.3_{1.0}}$ | $12.6_{1.3}$ | $\mathbf{2.4_{0.3}}$ | $\mathbf{3.6_{0.9}}$ |
| | Ckpt Ens | $\mathbf{4.2_{0.7}}$ | $14.2_{0.5}$ | $18.9_{0.2}$ | $\mathbf{7.2_{1.2}}$ | $15.7_{0.5}$ | $15.6_{0.3}$ |
| | LLLA | $22.4_{3.0}$ | $12.0_{1.5}$ | $8.2_{0.3}$ | $17.8_{1.8}$ | $6.7_{0.4}$ | $6.7_{0.7}$ |
| | LA | $14.2_{0.8}$ | $23.8_{1.0}$ | $28.3_{0.2}$ | $9.5_{2.8}$ | $17.3_{0.6}$ | $20.2_{0.5}$ |
| NLL ↓ | MAP | $2.64_{0.18}$ | $1.06_{0.06}$ | $0.53_{0.01}$ | $0.90_{0.04}$ | $0.42_{0.01}$ | $0.44_{0.01}$ |
| | MC Drop | $1.40_{0.06}$ | $0.53_{0.01}$ | $\mathbf{0.32_{0.02}}$ | $0.70_{0.02}$ | $\mathbf{0.34_{0.01}}$ | $\mathbf{0.38_{0.01}}$ |
| | Ckpt Ens | $\mathbf{0.62_{0.01}}$ | $\mathbf{0.47_{0.00}}$ | $0.42_{0.00}$ | $\mathbf{0.60_{0.01}}$ | $0.46_{0.00}$ | $0.47_{0.00}$ |
| | LLLA | $0.88_{0.05}$ | $0.66_{0.06}$ | $0.39_{0.02}$ | $0.82_{0.03}$ | $0.39_{0.01}$ | $0.41_{0.01}$ |
| | LA | $0.64_{0.01}$ | $0.55_{0.01}$ | $0.54_{0.01}$ | $0.62_{0.00}$ | $0.45_{0.01}$ | $0.50_{0.00}$ |

Figure 10: Fine-tuning of LlaMA2-7B across six common sense reasoning tasks (presented column-wise, with number of training examples in brackets), evaluated on the test set every 1000 gradient steps. Laplace (diagonal) prior precision is tuned using model evidence.

## J CODE IMPLEMENTATIONS

We open sourced an original implementation based on Laplace Redux (Daxberger et al., 2021a) and ASDL (Osawa et al., 2023) at https://github.com/adamxyang/laplace-lora, and a newer standalone implementation which we intended to support going forward at https://github.com/MaximeRobeyns/bayesian_lora.

Table 17: Comparison of different post-hoc methods applied to the fine-tuned LlaMA2-7B across six common sense reasoning tasks. Results are evaluated at the early stopping point of 5000 gradient steps.

| Methods | Metrics | WG-S | ARC-C | ARC-E | WG-M | OBQA | BoolQ |
|---|---|---|---|---|---|---|---|
| ACC ↑ | MAP | $67.4_{0.3}$ | $66.3_{0.6}$ | $84.7_{1.5}$ | $73.4_{0.4}$ | $78.7_{0.4}$ | $86.1_{0.2}$ |
| | MC Drop | $67.8_{0.1}$ | $65.3_{1.0}$ | $85.0_{1.3}$ | $73.2_{0.5}$ | $79.5_{0.2}$ | $86.0_{0.3}$ |
| | Ckpt Ens | $67.4_{0.2}$ | $65.5_{0.4}$ | $85.8_{0.2}$ | $73.6_{0.7}$ | $79.1_{0.1}$ | $86.3_{0.2}$ |
| | LLLA | $67.7_{0.3}$ | $65.5_{1.4}$ | $84.6_{1.2}$ | $73.6_{0.7}$ | $78.7_{0.7}$ | $86.0_{0.3}$ |
| | LA | $67.4_{0.4}$ | $63.2_{1.9}$ | $82.8_{0.2}$ | $73.5_{1.1}$ | $78.7_{0.4}$ | $86.1_{0.3}$ |
| ECE ↓ | MAP | $31.2_{0.3}$ | $31.0_{0.5}$ | $13.4_{1.3}$ | $23.0_{0.1}$ | $16.1_{0.6}$ | $4.0_{0.5}$ |
| | MC Drop | $29.4_{0.3}$ | $29.6_{0.8}$ | $12.4_{1.2}$ | $22.2_{0.2}$ | $15.0_{0.4}$ | $4.1_{0.4}$ |
| | Ckpt Ens | $29.7_{0.6}$ | $27.0_{1.5}$ | $\mathbf{9.8_{0.6}}$ | $17.4_{0.9}$ | $\mathbf{12.4_{0.3}}$ | $\mathbf{1.2_{0.4}}$ |
| | LLLA | $23.3_{1.7}$ | $20.9_{2.6}$ | $11.8_{1.9}$ | $22.5_{0.6}$ | $15.9_{0.3}$ | $4.1_{0.6}$ |
| | LA | $\mathbf{11.6_{0.5}}$ | $\mathbf{18.3_{0.5}}$ | $16.6_{3.7}$ | $\mathbf{6.6_{1.2}}$ | $17.2_{1.2}$ | $17.0_{0.9}$ |
| NLL ↓ | MAP | $3.15_{0.10}$ | $3.28_{0.29}$ | $1.26_{0.13}$ | $1.51_{0.05}$ | $0.99_{0.05}$ | $0.35_{0.01}$ |
| | MC Drop | $2.81_{0.11}$ | $2.82_{0.21}$ | $1.11_{0.10}$ | $1.41_{0.03}$ | $0.95_{0.04}$ | $0.35_{0.01}$ |
| | Ckpt Ens | $2.58_{0.15}$ | $2.36_{0.34}$ | $0.80_{0.06}$ | $0.87_{0.06}$ | $0.76_{0.01}$ | $\mathbf{0.33_{0.00}}$ |
| | LLLA | $1.02_{0.10}$ | $1.30_{0.11}$ | $0.90_{0.24}$ | $1.42_{0.05}$ | $0.96_{0.05}$ | $0.35_{0.01}$ |
| | LA | $\mathbf{0.64_{0.01}}$ | $\mathbf{1.00_{0.04}}$ | $\mathbf{0.57_{0.03}}$ | $\mathbf{0.55_{0.01}}$ | $\mathbf{0.69_{0.01}}$ | $0.44_{0.01}$ |

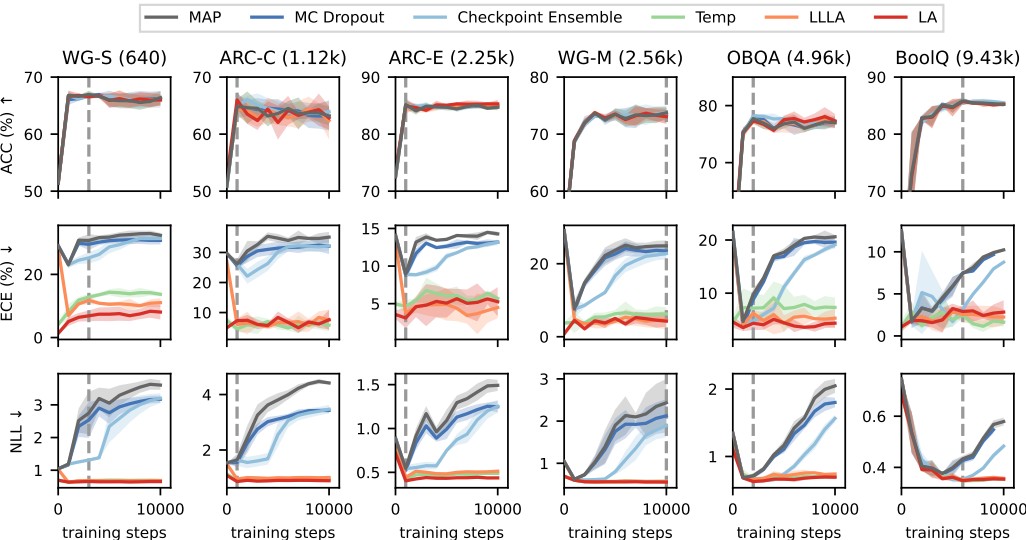

Figure 11: Fine-tuning of LlaMA2-7B across six common sense reasoning tasks (presented column-wise, with number of training examples in brackets), evaluated on the test set every 1000 gradient steps. The vertical dashed line gives the checkpoint with optimal MAP performance on a held-out validation set. Temperature scaling and Laplace (diagonal) prior precision are tuned on the validation set.

Table 18: Comparison of different post-hoc methods applied to the fine-tuned LlaMA2-7B across six common sense reasoning tasks, with a validation set split from the training set used for tuning temperature and Laplace prior precision. Results are evaluated at the best MAP performance checkpoint observed on the validation set.

| Methods | Metrics | WG-S | ARC-C | ARC-E | WG-M | OBQA | BoolQ |
|---|---|---|---|---|---|---|---|
| ACC ↑ | MAP | $67.0_{0.6}$ | $64.9_{1.1}$ | $85.2_{0.6}$ | $73.7_{0.9}$ | $77.7_{0.8}$ | $85.8_{0.4}$ |
| | MC Drop | $66.7_{0.3}$ | $64.9_{1.9}$ | $85.1_{0.5}$ | $73.5_{0.9}$ | $77.7_{0.2}$ | $85.9_{0.4}$ |
| | Ckpt Ens | $66.7_{0.3}$ | $64.9_{1.1}$ | $85.2_{0.6}$ | $73.8_{1.0}$ | $78.2_{0.2}$ | $85.4_{0.3}$ |
| | Temp | $67.0_{0.6}$ | $64.9_{1.1}$ | $85.2_{0.6}$ | $73.7_{0.9}$ | $77.7_{0.8}$ | $85.8_{0.4}$ |
| | LLLA | $66.7_{0.3}$ | $64.4_{1.0}$ | $85.1_{0.8}$ | $73.1_{1.2}$ | $77.2_{0.3}$ | $85.7_{0.4}$ |
| | LA | $66.5_{0.3}$ | $66.0_{1.2}$ | $85.0_{1.2}$ | $73.1_{1.1}$ | $77.3_{0.3}$ | $85.7_{0.5}$ |
| ECE ↓ | MAP | $30.8_{1.8}$ | $26.1_{1.4}$ | $8.9_{0.3}$ | $24.9_{1.3}$ | $9.8_{1.0}$ | $7.4_{0.1}$ |
| | MC Drop | $29.5_{1.6}$ | $25.6_{0.7}$ | $8.8_{0.6}$ | $23.5_{1.2}$ | $8.8_{0.8}$ | $7.5_{0.1}$ |
| | Ckpt Ens | $25.2_{1.6}$ | $26.1_{1.4}$ | $8.9_{0.3}$ | $22.8_{1.4}$ | $4.7_{0.5}$ | $3.2_{0.5}$ |
| | Temp | $12.8_{0.9}$ | $\mathbf{4.6_{1.0}}$ | $4.7_{0.8}$ | $6.3_{1.6}$ | $7.2_{2.6}$ | $\mathbf{2.5_{0.3}}$ |
| | LLLA | $11.8_{1.1}$ | $6.0_{1.8}$ | $3.7_{0.5}$ | $4.5_{2.2}$ | $6.2_{0.5}$ | $2.6_{0.9}$ |
| | LA | $\mathbf{6.9_{1.5}}$ | $7.3_{0.6}$ | $\mathbf{3.1_{1.2}}$ | $\mathbf{4.3_{1.3}}$ | $\mathbf{4.3_{1.1}}$ | $2.9_{0.5}$ |
| NLL ↓ | MAP | $2.75_{0.57}$ | $1.64_{0.19}$ | $0.54_{0.03}$ | $2.43_{0.50}$ | $0.71_{0.03}$ | $0.43_{0.01}$ |
| | MC Drop | $2.54_{0.49}$ | $1.55_{0.16}$ | $0.52_{0.04}$ | $2.12_{0.35}$ | $0.71_{0.04}$ | $0.43_{0.01}$ |
| | Ckpt Ens | $1.31_{0.04}$ | $1.64_{0.18}$ | $0.54_{0.03}$ | $1.89_{0.24}$ | $0.65_{0.02}$ | $\mathbf{0.35_{0.01}}$ |
| | Temp | $0.68_{0.01}$ | $0.90_{0.01}$ | $0.43_{0.02}$ | $0.58_{0.01}$ | $0.67_{0.02}$ | $\mathbf{0.35_{0.00}}$ |
| | LLLA | $0.68_{0.02}$ | $0.95_{0.03}$ | $0.44_{0.01}$ | $0.57_{0.01}$ | $0.66_{0.02}$ | $\mathbf{0.35_{0.00}}$ |
| | LA | $\mathbf{0.66_{0.02}}$ | $\mathbf{0.86_{0.03}}$ | $\mathbf{0.40_{0.02}}$ | $\mathbf{0.55_{0.01}}$ | $\mathbf{0.63_{0.01}}$ | $\mathbf{0.35_{0.01}}$ |

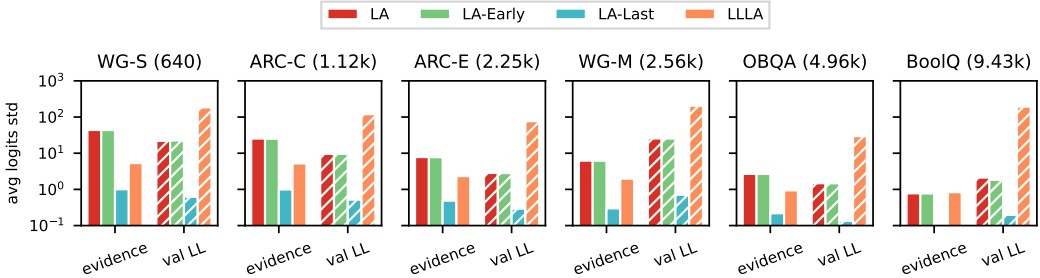

Figure 12: Bar chart of averaged logits standard deviation (bottom row) when optimizing Laplace model evidence (solid bars) on the training set at the early stopping checkpoint of 5000 training steps, and validation log-likelihood (dashed bars) on the validation set at the best MAP validation ACC checkpoint for each dataset. LA-Early involves taking a trained LA model, freezing the last layer and evaluating the uncertainty arising just from the "earlier" layers (i.e. all except the last layer). Likewise, LA-Last involves taking a trained LA model, freezing the earlier layers (i.e. all except the last layer) and evaluating the uncertainty arising just from the last layer. The ordering of the bars is the same as the ordering in the legend.

| Subject | Task |
|---|---|
| Computer Science (CS) | college computer science |
|  | computer security |
|  | high school computer science |
|  | machine learning |
| Engineering (Eng) | electrical engineering |
| Law | international law |
|  | jurisprudence |
|  | professional law |
| Health | anatomy |
|  | clinical knowledge |
|  | college medicine |
|  | human aging |
|  | nutrition |
|  | professional medicine |
|  | virology |

Table 19: MMLU subjects and tasks.

