# OpenReview forum: "Bayesian Low-rank Adaptation for Large Language Models"
_ICLR.cc/2024/Conference — ICLR 2024 poster_

### Official Review · Reviewer_CyBU · 2023-10-30

**Soundness:** 4 excellent
**Presentation:** 4 excellent
**Contribution:** 4 excellent
**Rating:** 8
**Confidence:** 3

**Summary:**

This paper proposed a new fine-tuning procedure based on LoRA using Bayesian prospective.

**Strengths:**

This paper is very well-written and the algorithm is presented in a very clean way. Although the algorithm is straightforward, as far as I know, nobody has made it work before. The empirical analysis is done carefully which is appreciated. Code is also linked.

**Weaknesses:**

1. It looks like all the tasks using in this paper are still classification tasks. How it is different from standard Bayesian deep learning evaluation? I want to understand where the benefit is coming from on those eval metrics through a principled way. I feel it will be good to comment some potential on generation tasks that can reflect more of the autoregressive nature of transformers.

2. Where is the comparison of computational complexity and memory usage? How much additional memory you will use?

**Questions:**

See above.

---

> ### Author Response · Authors · 2023-11-22
> **Response**
>
> Thank you for your constructive feedback and positive remarks on our work.
>
> > It looks like all the tasks using in this paper are still classification tasks. How it is different from standard Bayesian deep learning evaluation?
>
> We use classification tasks precisely so we can use standard, well-understood methods for assessing the quality of uncertainty estimation.
>
> > I want to understand where the benefit is coming from on those eval metrics through a principled way.
>
> The improvement on calibration metrics is arising because we use approximate Bayesian inference to estimate our uncertainty in fine-tuning weights given a very small number of fine-tuning examples.
>
> **Autoregressive generation**
>
> Given that this is the first investigation of Laplace for LoRA fine-tuning, we chose to focus on multiple-choice QA because that allowed us to use robust, well-understood calibrations metrics like ECE and NLL.
>
> As a next step, we are indeed excited by the possibility of investigating free-text generation: Laplace-LoRA certainly could be applied in this setting.  However, the development of robust, well-understood calibration metrics for free-text generation remains a challenging and open research question. Given the complexity of evaluating calibration in the free-text setting, this extension is out-of-scope here, and we leave it for future work.
>
> **Memory and runtime cost**
>
> We have added the memory and time cost comparison between LoRA fine-tuning and LoRA with post-hoc Laplace-LoRA in Section 5.4 and Table 5 in the main text.  The upshot is that the dominant additional cost arises from accumulating the low-rank Kronecker factors; that takes about 10\% of the time, and only slightly more memory (actually about 1-5\% more memory) than the original finetuning run.

---

### Official Review · Reviewer_sBHc · 2023-11-01

**Soundness:** 3 good
**Presentation:** 3 good
**Contribution:** 2 fair
**Rating:** 6
**Confidence:** 3

**Summary:**

This paper presented a Bayesian approach to improve the calibration of LoRA. It adopted Laplace approximation to the posterior of the low rank parameters. Empirical results on both in-distribution and out-distribution finetuning indicated the Bayesian approach does improve the calibration and the proposed Laplace approximation outperform a variety of baseline methods.

**Strengths:**

1. This paper is the first to present a comprehensive result on using Laplace approximation to LoRA for LLMs.
2. This paper has clear presentation with visualization.
3. Claims are supported with sufficient amount of convincing experiment result. e.g., smaller datasets experience larger difference in ECE compares with larger datasets.

**Weaknesses:**

1. Limited novelty. The Bayesian method part (as indicated in the paper) is well explored in the literatures listed in the software paper Laplace Redux [1]. This paper can be viewed as empirical results applying [1] to a specific model - LoRA for LLMs.
2. Majority of the benefits of Laplace-LoRA including ''post-hoc'' and ''scalable'' are from the existing method, which limits the contribution of this work. This one together with Weakness #1 above are the major concerns from my point of view.
3. There was some discussion on the cost to perform Laplace approximation when introducing Fasher. However, there is no empirical results that support it.
4. The ablation study (LLLA vs LA) is also from the [1]. Only comparing LA with Last Layer LA is not very convincing on where the uncertainty comes from.

[1] Erik Daxberger, Agustinus Kristiadi, Alexander Immer, Runa Eschenhagen, Matthias Bauer, and Philipp Hennig. Laplace redux-effortless bayesian deep learning. NeurIPS, 2021.

**Questions:**

1. Laplace Redux has discussion on relative wall-clock time cost and memory consumption comparison with other methods, is there similar result for Laplace-LoRA?
2. More ablation study is expected. For example, LA on different types of layers (attention, dense, etc.), LA on different layers and different number of layers (first k layers or random select layers), or some fusion of the two mentioned above.

---

> ### Author Response · Authors · 2023-11-22
> **Response**
>
> Thanks for your constructive, thoughtful feedback.
>
> **Novelty**
>
> Bayesian deep learning is rarely if ever used by LLM practitioners.  That's because there are currently no effective and efficient methods for Bayesian finetuning of modern LLMs.  Here, we show comprehensively that highly efficient Bayesian methods can dramatically improve calibration in modern LLMs.  That's the key novelty in this paper, and it opens the way to practical applications of Bayesian methods in modern LLMs, along with a far more thorough exploration of Bayesian methods for LLMs in future work.
>
> Regarding the novelty of our Laplace-LoRA approach, we believe its significance lies in the innovative integration of well-studied methods to create a novel framework for Bayesian fine-tuning on LLMs. This integration is not trivial and addresses a critical gap in current methodologies. In addition, we opensourced code that works with any publicly available LLMs on Huggingface.
>
> **Ablations**
>
> Laplace Redux only compares LLLA vs LA for ResNets for simple computer vision tasks such as CIFAR-10.  This tells us very little about what to expect in modern LLM finetuning.  Indeed, we come to the opposite conclusion from the Laplace Redux paper.  They find that LLLA is sufficient in their tasks.  While we find that retaining uncertainty at all layers is important in parameter-efficient LLM finetuning with low-rank adaptation.
>
> Additionally, our experiments weren't really directed at finding "where the uncertainty comes from".  They were directed at discovering why, when optimizing the prior using the Laplace estimate of the marginal likelihood, the uncertainties for LLLA and LA were so different.  Specifically, the uncertainty for LLLA was far too low, whereas the uncertainty for LA was about right.  Our conclusion was that there is little uncertainty arising at the last layer.
>
> In response to your request, we expanded our analysis in Appendix G.3, by applying LA to different LoRA layers. Specifically, we experimented with applying LA on the first 8, 16, 24, and 32 (all) layers. While we observed only small differences, there does seem to be a small improvement in the NLL when LA is applied beyond the top 8 layers.
>
> **Memory and runtime cost**
>
> We have added the memory and time cost comparison between LoRA fine-tuning and LoRA with post-hoc Laplace-LoRA in Section 5.4 and Table 5 in the main text.  The upshot is that the dominant additional cost arises from accumulating the low-rank Kronecker factors; that takes about 10\% of the time, and only slightly more memory memory (actually about 1-5\% more memory) than the original finetuning run.

---

> > ### Comment · Reviewer_sBHc · 2023-11-23
> >
> > I appreciate the author's rebuttal and their response to my question. I have increased my score.

---

### Official Review · Reviewer_5qzU · 2023-11-06

**Soundness:** 3 good
**Presentation:** 4 excellent
**Contribution:** 3 good
**Rating:** 6
**Confidence:** 4

**Summary:**

This paper introduces Laplace-LoRA, a Bayesian inference method designed specifically for the LoRA parameters in LLM fine-tuning using a post-hoc Laplace approximation. They conducted extensive experiments on six commonsense reasoning tasks and show dramatic improvements in calibration of fine-tuned LLMs.

**Strengths:**

1. This idea of combining Laplace inference with the fine-tuning LLMs using LoRA adapters is novel, which provides a new way of doing Bayesian fine-tuning on LLMs.
2. They conducted extensive experiments on six commonsense reasoning tasks under in/out-of-distribution settings and provided detailed analysis of the experiment results.
3. The writing is well-structured, clear and easy to understand.

**Weaknesses:**

1. It has some novelty, but not dramatic, because both Laplace Approximation and LoRA method are well-studied.
2. It is quite weird that the Section 3 Background followed by Section 4 Results directly, without a Method section in between. Maybe it needs a better section name.

**Questions:**

Does free-text generation tasks (e.g., free-from QA) still work under this framework? Why that kind of reasoning tasks are not considered in your experiments?

---

> ### Author Response · Authors · 2023-11-22
> **Response**
>
> Thank you for your constructive feedback and positive remarks on our work.
>
> **Novelty**
>
> Bayesian deep learning is rarely if ever used by LLM practitioners.  That's because there are currently no effective and efficient methods for Bayesian finetuning of modern LLMs.  Here, we show comprehensively that highly efficient Bayesian methods can dramatically improve calibration in modern LLMs.  That's the key novelty in this paper, and it opens the way to practical applications of Bayesian methods in modern LLMs, along with a far more thorough exploration of Bayesian methods for LLMs in future work.
>
> Regarding the novelty of our Laplace-LoRA approach, we believe its significance lies in the innovative integration of well-studied methods to create a novel framework for Bayesian fine-tuning on LLMs. This integration is not trivial and addresses a critical gap in current methodologies. In addition, we opensourced code that works with any publicly available LLMs on Huggingface.
>
> **Methods**
>
> We have added a Methods section describing the nontrivial methods required for the low-rank K-FAC approach that we designed for Laplace-LoRA, with detailed derivations in Appendix E.
>
> **Autoregressive generation**
>
> Given that this is the first investigation of Laplace for LoRA fine-tuning, we chose to focus on multiple-choice QA because that allowed us to use robust, well-understood calibrations metrics like ECE and NLL.
>
> As a next step, we are indeed excited by the possibility of investigating free-text generation: Laplace-LoRA certainly could be applied in this setting.  However, the development of robust, well-understood calibration metrics for free-text generation remains a challenging and open research question. Given the complexity of evaluating calibration in the free-text setting, this extension is out-of-scope here, and we leave it for future work.

---

> ### Comment · Reviewer_5qzU · 2023-11-23
>
> Thanks for the clarification! My concerns have been addressed.

---

### Official Review · Reviewer_EVj9 · 2023-11-12

**Soundness:** 3 good
**Presentation:** 4 excellent
**Contribution:** 3 good
**Rating:** 6
**Confidence:** 4

**Summary:**

A Bayesian LoRA approach has been proposed to fine-tune large language models (LMMs) by estimating the weight posterior through layer-wise Laplace approximation. Experiment results on six QA benchmark datasets were provided in terms of accuracy, calibration, and OOD generalization, compared with three baseline methods.

**Strengths:**

- **Timely research**: The proposed method focuses on improving the calibration performance when finetuning LLMs on small-scale datasets, which is an important and urgent research problem along with the rapid growth of large models.

- **Clear Bayesian treatment**: The proposed method adopts well-established techniques from prior works of Bayesian neural networks and uncertainty reasoning, and successfully incorporates such a Bayesian treatment into parameter-efficient tuning approaches. The proposed Laplace-LoRA seems to be a scalable solution to enable uncertainty estimation for large models.

- **Good experiment design**: Despite some practical issues (see *weakness*), the experiment was conducted well on six public datasets with several strong baseline methods, in terms of three settings -- 1) early stopping, 2) finetuning with validation, and 3) OOD generalization. The proposed method generally achieves better ECE scores across different cases.

**Weaknesses:**

- **Unclear uncertainty estimation**: While the proposed Laplace-LoRA naturally estimates the weight posterior, it is unclear how to apply the proposed method to compute model uncertainties. Also, it remains unclear if the proposed method can handle the structured uncertainty estimation for next-token predictions (e.g., *Uncertainty estimation in autoregressive structured prediction, ICLR'21*). It would also be interesting to compare the proposed method with semantic uncertainty [Kuhn et al., ICLR'23].
- **Lack of LLM backbones**: The empirical evidence of the proposed method is somewhat weak due to the lack of more LLM backbones. Some concerns include: 1) can the proposed Laplace-LoRA work with larger model sizes (e.g., 13B, 70B)? 2) Can Laplace-LoRA be applied to models other than LLaMA2 (e.g., Mistral 7B, GPT-2, etc.)?
- **Comparison to ensemble**: Besides the checkpoint ensemble, it is also expected to implement a baseline given by the LoRA ensemble (similar to the deep ensemble approach).

**Questions:**

Please refer to the questions raised in *Weaknesses*.

---

> ### Author Response · Authors · 2023-11-22
> **Response**
>
> Thanks for your careful and thoughtful response!
>
> **Model uncertainties**
>
> In common with with the standard Laplace approach, we compute a Gaussian approximation to the distribution over logits implied by the uncertainty in weights.
> We do this by linearising the model (Eq. 12, 13 and 15).
>
> **Autoregressive generation**
>
> Given that this is the first investigation of Laplace for LoRA fine-tuning, we chose to focus on multiple-choice QA because that allowed us to use robust, well-understood calibrations metrics like ECE and NLL.
>
> As a next step, we are indeed excited by the possibility of investigating free-text generation: Laplace-LoRA certainly could be applied in this setting.  However, the development of robust, well-understood calibration metrics for free-text generation remains a challenging and open research question. Given the complexity of evaluating calibration in the free-text setting, this extension is out-of-scope here, and we leave it for future work.
>
> **Backbones**
>
> The approach can be applied to any setting where you would use a LoRA adapter.  We had experiments in the original manuscript on RoBERTa-base and RoBERTa-large  (Figs. 4-7, Tables 9-12 in Appendix G1). We have added additional experiments with Mistral-7B (Table 13 in Appendix G3).  All these experiments show the same patterns as those in the main text.  We will to add more extensive experiments on Mistral, as well as experiments on Llama 13B models for the camera-ready (we had intended to include them here, but we ran out of time in the discussion period).
>
> **Ensemble baseline**
>
> We have added LoRA ensemble to Fig. 1 (light green line) and Table 1 and 3.  Ensembles give surprisingly little benefit in this setting, which is broadly in-line with another ICLR submission (number 3048) examining precisely this question.

---

### Author Response · Authors · 2023-11-22
**Overall response**

In this work, we introduce perhaps the first scalable Bayesian method for finetuning in modern LLMs: Laplace LoRA.  We show that the resulting method offers dramatic benefits in terms of calibration.  Our work thus opens the way to practical applications of Bayesian methods in modern LLMs, along with a far more thorough exploration of Bayesian methods for LLMs in future work.


The reviewers recognised the timeliness, novelty and clarity of the contribution, along with the strength of our experimental results.

**Timeliness:**
* "The proposed method focuses on improving the calibration performance when finetuning LLMs on small-scale datasets, which is an important and urgent research problem along with the rapid growth of large models." (R1)

**Novelty:**
* "This idea of combining Laplace inference with the fine-tuning LLMs using LoRA adapters is novel" (R2)
* "This paper is the first to present a comprehensive result on using Laplace approximation to LoRA for LLMs."(R3)
* "nobody has made it work before."(R4)

**Clarity:**
* "The writing is well-structured, clear and easy to understand." (R2)
* "This paper has clear presentation with visualization." (R3)
* "very well-written" (R4)

**Experiments:**
* "Good experiment design" (R1)
* "extensive experiments" (R2)
* "convincing experiment result" (R3)
* "The empirical analysis is done carefully" (R4)

---

### Meta-Review · Area_Chair_CGCF · 2023-12-05

**Metareview:**

(a)  A Bayesian LoRA approach has been proposed to fine-tune large language models (LMMs) by estimating the weight posterior through layer-wise Laplace approximation. Experiment results on six QA benchmark datasets were provided in terms of accuracy, calibration, and OOD generalization, compared with three baseline methods.
(b)  The idea of combining Laplace inference with the fine-tuning LLMs using LoRA adapters is novel, which provides a new way of doing Bayesian fine-tuning on LLMs.   They conducted extensive experiments on six commonsense reasoning tasks under in/out-of-distribution settings and provided detailed analysis of the experiment results.  The writing is well-structured, clear and easy to understand.  Good rebuttals.
(c)  Should explore different text tasks.  Some improvements to writeup suggested.

**Justification For Why Not Higher Score:**

Only one glowing substantial review, others were marginal or short.

**Justification For Why Not Lower Score:**

Unanimous decision.

---

### Decision · Program_Chairs · 2024-01-16

Accept (poster)